# Learning What Matters: Prioritized Concept Learning via Relative Error-driven Sample Selection

## Abstract

Instruction tuning has been central to the success of recent vision-language models (VLMs), but it remains expensive—requiring large scale datasets, high-quality annotations and large-compute budget. We propose **PR**ioritized c**O**ncept learnin**G** via **R**elative **E**rror-driven **S**ample **S**election – **PROGRESS** – a data- and compute-efficient framework that enables VLMs to dynamically select what to learn next based on their evolving needs during training. At each stage, the model tracks its learning progress across skills and selects the most *informative* samples: those it *has not already mastered* and *are not too difficult to learn* at the current state of training. This strategy effectively controls skill acquisition and the order in which skills are learned. Specifically, we sample from skills showing the highest learning progress, prioritizing those with the most rapid improvement. Unlike prior methods, PROGRESS requires no upfront answer annotations, querying answers only on a *need basis*, avoids reliance on additional supervision from auxiliary VLM, or compute-heavy gradient computations for data selection. Experiments across multiple instruction-tuning datasets of varying scales demonstrate that PROGRESS consistently outperforms state-of-the-art baselines with much less data and supervision. Additionally, we show strong cross-architecture generalization to different VLMs and transferability to larger models, validating PROGRESS as a scalable solution for efficient learning.[1]

## 1 Introduction

Multimodal vision-language models (VLMs) such as GPT-4V (OpenAI et al., 2024), Gemini (Team et al., 2023), LLaVA (Liu et al., 2023b;a), and InternVL (Chen et al., 2024b) demonstrate impressive general-purpose capabilities across tasks like image comprehension and visual question answering. Much of this success stems from large-scale fine-tuning on high-quality image-text corpora, particularly visual instruction-tuning (IT) datasets (Zhang et al., 2023; Xu et al., 2024), which significantly enhance instruction-following and reasoning abilities. A growing trend in building stronger VLMs has been to simply scale up: collecting larger more diverse IT datasets with better annotations and using them to instruction-tune increasingly powerful models (Chen et al., 2023; Liu et al., 2023a).

However, such pipelines are increasingly resource-intensive—annotation-heavy when relying on human-labeled supervision (*e.g.*, bounding boxes, object tags) and monetarily costly when generating instructions via proprietary models like GPT-4 (Liu et al., 2023b;a), alongside significant computational overhead. These factors make such pipelines increasingly inaccessible to individual researchers and smaller academic labs. More importantly, it is unclear whether the entirety of these large corpora is necessary for strong VLM performance. We posit that many samples are redundant or uninformative, and that comparable results could be achieved using fewer, informative samples.

To this end, we investigate how to select the *most informative* visual instruction-tuning (IT) samples based on the model's own evolving learning state. We ask: *Can VLMs indicate what they can most effectively learn at a give stage of training?* Inspired by curriculum learning, we develop a framework in which the model periodically self-evaluates its current knowledge and identifies the skills it is

---

[1]Code will be released on publication.

ready to acquire next—those that would most benefit its learning progress. Specifically, we track the relative change in skill performance across iterations to estimate where learning improves fastest, encouraging the model to prioritize these skills. We hypothesize that this enables the VLM to actively select training samples that are most informative: those that are *not already mastered* by the model, and are *not too difficult* for the model to learn at its current stage. Overall, PROGRESS is designed to adapt to the model's evolving learning state by helping it acquire essential skills, while also promoting diversity across selected concepts—a property crucial for capturing important modes in the data and supporting generalization.

Experimental results across multiple instruction-tuning datasets of varying scale demonstrate that PROGRESS achieves up to **99–100%** of the full-data performance while using only **16–20%** of the labeled training data. In addition to these gains, PROGRESS offers several practical advantages over existing approaches. First, unlike static scoring-based methods(Paul et al., 2021; Coleman et al., 2019; Marion et al., 2023; Hessel et al., 2022) or concept-driven strategies that rely on additional reference VLMs (Lee et al., 2024), our method uses dynamic feedback from the model's own learning progress to guide training. Second, while many prior methods assume full access to ground-truth annotations upfront (Lee et al., 2024; Wu et al., 2025), we operate in a more realistic setting where the training pool is initially unlabeled and answers are queried only on a *need basis*—drastically reducing annotation cost. Third, instead of merely selecting *which* samples to train on (Lee et al., 2024; Wu et al., 2025), our approach also decides *when* to introduce each skill—enabling curriculum-style control over both skill acquisition and learning order. Our contributions are as follows:

- We propose **PROGRESS**, a dynamic, progress-driven framework for selecting the most informative samples during VLM instruction tuning—based on relative improvement across automatically discovered skills.

- Our method achieves near full-data performance using only 16–20% supervision across multiple instruction-tuning datasets of varying scale and across different VLMs—including the widely used LLaVA-v1.5-7B. It generalizes well across architectures, showing strong results on larger-capacity models like LLaVA-v1.5-13B and newer designs such as Qwen2-VL, while consistently outperforming competitive baselines and prior data-efficient methods.

- We analyze *what* skills the model prioritizes and *when*, revealing an interesting curriculum over skill types and difficulty—offering new insights into efficient VLM training.

## 2 RELATED WORK

**Data Efficient Learning for VLMs.** Prior approaches to efficient VLM training (see Fig. 1) typically select a coreset—*i.e.*, a representative subset of the data for training—using static metrics such as CLIP-Score (Hessel et al., 2022), EL2N (Paul et al., 2021), perplexity (Marion et al., 2023), entropy (Coleman et al., 2019), or by training auxiliary scoring networks (Chen et al., 2024a). However, these methods perform one-time sample selection before training and cannot adapt to the model's evolving needs. Moreover, static score metrics often miss important data modes, leading to poor diversity and reduced generalization (Lee et al., 2024; Maharana et al., 2025)—in some cases even underperforming random selection (Lee et al., 2024). Gradient-based methods (Wu et al., 2025; Liu et al., 2024c), while more principled, are computationally prohibitive—requiring large memory to store high-dimensional gradients and hundreds of GPU hours (e.g., ICONS[2] (Wu et al., 2025))—contradicting the very goal of efficient training. Some also assume access to explicit knowledge of the target task or distribution through labeled samples, as in ICONS (Wu et al., 2025), which is rarely realistic in general-purpose VLM settings. More recently, prominent work has explored selection using additional reference VLMs—auxiliary models that themselves require large-scale instruction tuning. A notable example is COINCIDE (Lee et al., 2024), which extracts internal activations from a separately trained VLM (e.g., TinyLLaVA (Zhou et al., 2024)) to guide coreset selection. However, COINCIDE exhibits several limitations: it requires a fully trained additional auxiliary VLM, performs static one-time selection without controlling the order of skills being learned, needs ground-truth annotations for the entire dataset, and requires manual human inspection to first select appropriate activations—all of which make it resource-intensive and difficult to scale.

**Key Differences-** In contrast, our method brings together the best of all worlds (summarized in Fig. 1): Notably, different from COINCIDE, our method selects and annotates samples dynamically adapting to the model's learning state (*row 1*), effectively controlling skills and order of aquisition

---

[2]Unpublished concurrent work; we reproduce and compare in Appendix A.4

(*row 2*), require no additional reference VLMs, no human-in-the-loop decisions (*row 3*), and needs supervision strictly on a *need basis* to only 20% of the dataset (*row 7*). Furthermore, unlike prior methods (Fig. 1), our approach requires no explicit knowledge of the target task or distribution (*row 4*), avoids compute-heavy gradient computation (*row 5*), and promotes diverse skill coverage (*row 6*), making it a practical solution for efficient and scalable VLM training in real-world.

**Curriculum Learning, Self-Paced Learning, and Active Learning.** Curriculum learning improves generalization by ordering data from easy to hard (Bengio et al., 2009), while self-paced extensions adapt this order to the learner's own progress (Kumar et al., 2010; Sachan & Xing, 2016). These ideas have been explored in NLP (Sachan & Xing, 2016; Mindermann et al., 2022) and in controlled multimodal settings such as CLEVR (Misra et al., 2017), but typically on small-scale models with external heuristics. In contrast, PROGRESS scales these principles to real-world VLM instruction tuning by selecting informative samples at the *skill* level using unsupervised clustering and dynamic

| Criterion \ Method | Random | Perplexity | EL2N | Sem-DeDup | Self-Filter | ICONS* | COINCIDE | PROGRESS |
|---|---|---|---|---|---|---|---|---|
| 1. Dynamic Selection | ✗ | ✗ | ✗ | ✗ | ✗ | ✗ | ✗ | ✓ |
| 2. Order of Skills | ✗ | ✗ | ✗ | ✗ | ✗ | ✗ | ✗ | ✓ |
| 3. Additional VLM Access | ✗ | ✗ | ✓ | ✓ | ✓ | ✗ | ✓ | ✗ |
| 4. Target Task Access | ✗ | ✗ | ✗ | ✗ | ✗ | ✓ | ✗ | ✗ |
| 5. Heavy-Gradient Overhead | ✗ | ✗ | ✗ | ✗ | ✗ | ✓ | ✗ | ✗ |
| 6. Diversity of Skills | ✓ | ✗ | ✗ | ✓ | ✗ | ✓ | ✓ | ✓ |
| 7. Annotation Budget | 20 % | 20 % | 100 % | 100 % | 100 % | 100 % | 100 % | 20 % |
| 8. Training Budget | 20 % | 20 % | 20 % | 20 % | 20 % | 20 % | 20 % | 20 % |

Figure 1: **Comparison with Prior Efficient Learning Methods for VLMs. Green** denote **desirable** properties for efficient learning, while **Red** indicate **limitations**. PROGRESS *satisfies all key desirable criteria* while requiring only *20% data*. See Appendix A.1 for details of prior approaches.

progress signals. Although related to active learning (AL), which reduces annotation cost via uncertainty-based sampling, PROGRESS instead prioritizes skills with the highest relative improvement—yielding a curriculum-like ordering rather than static uncertainty-driven selection. For completeness, we compare against strong AL methods (Table 1) and curriculum strategies (Table 3).

## 3 PROBLEM SETTING AND OVERALL FRAMEWORK

**Problem Setting.** We now formally introduce the data-efficient learning setting for training VLMs. We denote an image by $I$, a question by $Q$, forming an image-question pair $(I, Q) \in \mathbb{U}$, where $\mathbb{U}$ is an unlabeled pool of such pairs. Unlike previous efficient learning methods, we do not assume access to the corresponding answers $A \in \mathbb{A}$ for all pairs in $\mathbb{U}$, and thus refer to this pool as unlabeled. The learner is provided with: (1) the unlabeled pool $\mathbb{U}$; and (2) a fixed answer budget $b$, specifying the maximum number of pairs from $\mathbb{U}$ for which it can query an answer $A \in \mathbb{A}$ and use for training, where $|\mathbb{A}| = b \ll |\mathbb{U}|$. The goal is to learn a vision-language model $\texttt{VLM}(A \mid I, Q)$ that can accurately predict an answer for a new image-question pair, while only using $b$ selected and labeled samples during training. The central challenge lies in identifying the *most informative* $(I, Q)$ pairs to annotate within the constrained budget $b$, such that the resulting model trained on these $(I, Q, A)$ pairs performs comparably to one trained on the fully labeled dataset. This setup *follows standard setting* used in prior data-efficient learning (Lee et al., 2024; Wu et al., 2025; Chen et al., 2024a)

**Overall Framework.** Our overall framework for efficient training of VLMs is shown in Figure 2. We employ a two-stage pipeline:

(1) Multimodal Concept Categorization. Given an unlabeled data pool $\mathbb{U}$ containing image-question pairs $(I, Q) \in \mathbb{U}$, we first partition $\mathbb{U}$ into $K$ skill clusters in a *fully unsupervised* manner, assigning each sample $(I, Q)$ to a specific skill. This categorization enables tracking the model's progress on individual skills and supports a self-paced training strategy where the model's own learning signals determine which skills to prioritize next.

(2) Prioritized Concept Learning. During training, the model periodically self-evaluates its knowledge by comparing its current performance to prior state, identifying skills where performance improves fastest relative to prior state. Samples $(I, Q)$ from these skills are then selected and answer annotations $A \in \mathbb{A}$ are queried only for these selected samples.

Overall, our model dynamically selects diverse, informative samples throughout training, in alignment with its evolving learning needs. To ensure that skill-level performance estimates are credible at the start of training—when the model is still untrained—we begin with a brief *warmup phase* consistent with prior work (see details in Appendix A.3). This warmup, along with subsequent samples selected through our prioritized strategy, together make up the total sample budget of $b$, ensuring that training data in the form of $(I, Q, A)$ **never exceeds the specified budget**. We validate that using the warmup set alone to train the model yields significantly worse performance compared to our proposed strategy—highlighting the importance of progress-driven sampling (see Table 3 row

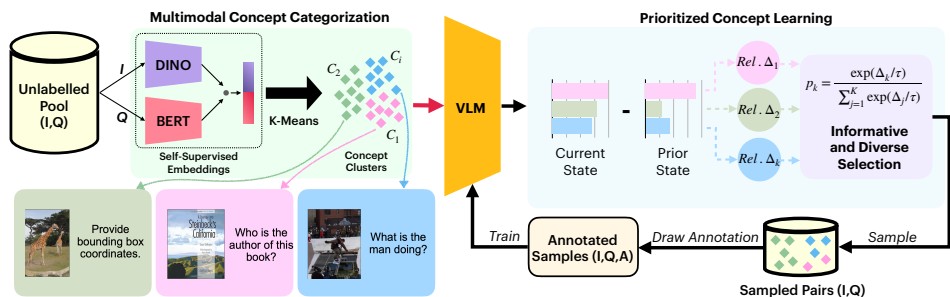

Figure 2: **Overall Pipeline.** Our framework consists of two stages: (1) *Multimodal Concept Categorization*, which partitions the unlabeled pool $\mathbb{U}$ into distinct skills by assigning each sample $(I, Q)$ to a specific skill cluster, and (2) *Prioritized Concept Learning*, where the model actively selects the most informative samples—those showing the highest improvement in its objective (e.g., accuracy or loss) relative to its prior state. We query annotations for only these selected samples on a *need basis*, forming the labeled set $(I, Q, A)$, which is used for training.

1). Our framework trains model with much less data, supervision and time, while controlling both skill acquisition and learning order.

## 3.1 MULTIMODAL CONCEPT CATEGORIZATION

We begin by identifying diverse skills from the unlabeled data pool through a fully ***unsupervised concept categorization*** module that partitions $\mathbb{U}$ into $K$ skill clusters using spherical k-means.

Each sample $(I, Q) \in \mathbb{U}$ is assigned to a cluster based on cosine similarity from multimodal concatenated self-supervised DINO (Oquab et al., 2024) (for image $I$) and BERT (Devlin et al., 2019) (for text question $Q$) features. Jointly leveraging both modalities yields purer clusters with higher intra-cluster and lower inter-cluster similarity compared to unimodal partitioning (see Fig. 3, Appendix C.1)—enabling accurate tracking of skill-level progress during training. Unlike COIN-CIDE (Lee et al., 2024),(closest best performing prior work) which requires activations from a additional VLM for skill categorization, ground-truth annotations for entire dataset, and human inspection of activations; our categorization is **fully unsupervised**—requiring no annotations, auxiliary models, or manual intervention.

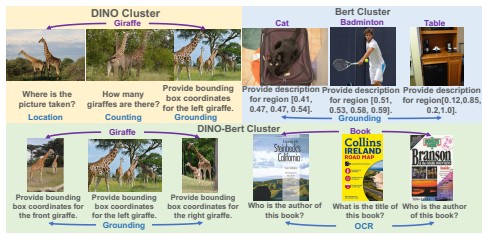

Figure 3: **Cluster Visualization.** Clustering with multimodal DINO-BERT features ensures purer skill clusters with higher intra-cluster and lower inter-cluster similarity compared to uni-modal partitioning. See **Word Cloud Visualization** Appendix C.1.

## 3.2 PRIORITIZED CONCEPT LEARNING: CAN VLMS INDICATE WHAT THEY CAN MOST EFFECTIVELY LEARN AT A GIVE STAGE OF TRAINING?

Our goal is to guide the VLM to prioritize skills it can most readily learn and improve upon. Since human intuition about task difficulty may not align with model's difficulty in its feature and hypothesis space (Sachan & Xing, 2016), we adopt a self-paced strategy where the model's own learning progress determines what to learn next. Inspired by curriculum learning (Kumar et al., 2010; Sachan & Xing, 2016), we select the most informative samples—those that yield the greatest improvement in the model's objective (*e.g.*, **accuracy or loss**) relative to its prior state.

Formally, given an unlabeled pool $\mathbb{U} = \{(I, Q)\}$ partitioned into skill clusters $\mathcal{C} = \{C_1, C_2, \ldots, C_K\}$, we define model's learning state at step $t$ by its accuracy $\text{Acc}_k^{(t)}$ on each skill-cluster $k$, computed over the ***training*** **data seen** by model so far. The *relative change in performance* across steps quantifies learning progress per skill, which is used to guide sample selection. We compute the expected accuracy improvement for each skill cluster between step $t$ and $t - \gamma$:

$$\Delta_k = \frac{\text{Acc}_k^{(t)} - \text{Acc}_k^{(t-\gamma)}}{\text{Acc}_k^{(t-\gamma)} + \epsilon} \tag{1}$$

where $\epsilon$ ensures numerical stability. The score $\Delta_k$ captures how rapidly the model is improving on skill cluster $k$, serving as a proxy for sample *informativeness*. By prioritizing high $\Delta_k$ clusters , the model focuses on skills it can improve on most rapidly—*thereby enforcing a self-paced curriculum that dynamically adapts to the model's learning state* (Sachan & Xing, 2016)—controlling both the acquisition of skills and the order in which they are learned. In addition to the selection strategy in Eqn 1, we sample $\delta\%$ of points at random to encourage exploration of new or underrepresented skills in dataset, following prior curriculum learning work (Kumar et al., 2010; Misra et al., 2017).

Annotations are queried only for selected samples forming the labeled set $(I, Q, A)$ for training. This *need-based annotation* strategy avoids the costly requirement of full supervision used in prior coreset methods (such as COINCIDE (Lee et al., 2024)), offering more scalable & efficient training.

However, naively selecting samples from only the highest-improvement cluster can hurt diversity by concentrating on a narrow skill set and leading to mode collapse—an issue known to degrade performance in prior work (Lee et al., 2024). To mitigate this, we propose to sample from multiple high $\Delta_k$ clusters in proportion to their relative improvement using a *temperature-controlled softmax*:

$$p_k = \frac{\exp(\Delta_k/\tau)}{\sum_{j=1}^{K} \exp(\Delta_j/\tau)} \tag{2}$$

Here, $p_k$ is the probability of sampling from cluster $k$, and $\tau$ controls the sharpness of the distribution. Lower $\tau$ emphasizes top clusters but risks mode collapse by repeatedly sampling from a narrow skill set (higher informativeness, lower diversity); higher $\tau$ promotes broader sampling and better skill coverage. This balance between **informativeness and diversity** is critical for effective and robust learning (see ablation Fig. 5). The sampling budget at given step $t$ is then allocated proportionally to $p_k$, and only the selected samples are annotated as $(I, Q, A)$ triplets for training.

# 4 EXPERIMENTS AND RESULTS

## 4.1 EXPERIMENTAL SETUP

**Datasets and Models.** To demonstrate effectiveness and generalizability across different scales of instruction-tuning (IT) data, we follow *standard protocol* (Lee et al., 2024) & conduct experiments on two IT datasets: large-scale LLaVA-665K (Liu et al., 2023b) containing $\sim 0.6$ **Million samples** & Vision-Flan-191K (Xu et al., 2024). For target VLMs, we primarily use **LLaVA-v1.5-7B** (Liu et al., 2023b), following prior work, and additionally report results on **LLaVA-v1.5-13B** (Liu et al., 2023b) to test scalability, and **Qwen2-VL-7B** (Wang et al., 2024) and **Qwen2.5-VL-32B-Instruct** (Bai et al., 2025) (See Appendix B.1) to test generalization to newer, larger architectures.

**Implementation Details.** Following standard protocol (Lee et al., 2024; Wu et al., 2025), we start from a pretrained model and fine-tune it on the instruction-tuning dataset using LoRA (Hu et al., 2021) with the official hyperparameters from LLaVA-1.5. For the accuracy variant, cluster-wise accuracy is estimated with an LLM judge that compares VLM outputs to ground-truth answers, though this step is not required for our loss-based variant. We *strictly follows standard* setup, evaluation protocols and metrics from prior work to ensure consistency and fair comparison (Lee et al., 2024; Wu et al., 2025). Additional implementation details are provided in Appendix A.2.

**Baselines.** Consistent with previous work (Lee et al., 2024), we compare PROGRESS against strong baselines spanning five major categories: (1) scoring function methods (CLIP-Score, EL2N(Paul et al., 2021), Perplexity (Marion et al., 2023)); (2) deduplication-based selection (SemDeDup (Abbas et al., 2023)); (3) graph-based methods (D2-Pruning (Maharana et al., 2024)); (4) external curriculum driven (Self-Filter (Chen et al., 2024a)); and (5) concept-diversity approaches (COINCIDE (Lee et al., 2024), Self-Sup (Sorscher et al., 2022)). Our baselines span both active learning (denoted by **AL**) and curriculum learning methods (denoted by **CL**) for comprehensive comparison. Following previous work, we include *Random*—a competitive baseline shown to perform well due to its diversity—and *Full-Finetune*, representing the performance upper bound with full data/supervision training. Details for baselines follow previous work (Lee et al., 2024) (See Appendix A.1 for further details).

**Evaluation Benchmark.** Following prior work, we evaluate our approach on a diverse suite of 14 benchmarks with wide coverage of skills: perceptual reasoning (VQAv2 (Goyal et al., 2017), VizWiz (Gurari et al., 2018)), textual reasoning (TextVQA (Singh et al., 2019)), compositional reasoning (GQA (Hudson & Manning, 2019)), object hallucinations (POPE (Li et al., 2023b)), multilingual understanding (MMBench-cn (Liu et al., 2024b), CMMMU (Ge et al., 2024)), instruction-following

Table 1: Comparison of coreset selection techniques for training LLaVA-v1.5 on the LLaVA-665K dataset using 20% sampling ratio.[3] *AL Only* refers to methods which use active learning principles. *AL & CL* refers to methods which use combination of active and curriculum learning principles. Methods highlighted in orange require **additional** reference VLMs and 100% dataset annotations for coreset selection, while methods highlighted in light green do not require either. The benchmark results are highlighted with best and second best within the respective categories (i.e, with and without utilizing additional information). The best and the second best relative score are in **bold** and underlined, respectively. Reported numbers are averaged over 3 runs.

| Method | VQAv2 | GQA | VizWiz | SQA-I | TextVQA | POPE | MME | MMBench en | MMBench cn | LLaVA-Wild | SEED | AI2D | ChartQA | CMMMU | Rel. (%) |
|---|---|---|---|---|---|---|---|---|---|---|---|---|---|---|---|
| **LLaVA-v1.5-7B** | | | | | | | | | | | | | | | |
| 0  Full-Finetune | 79.1 | 63.0 | 47.8 | 68.4 | 58.2 | 86.4 | 1476.9 | 66.1 | 58.9 | 67.9 | 67.0 | 56.4 | 16.4 | 22.1 | 100 |
| 1  Self-Sup (*AL Only*) | 74.9 | 59.5 | 46.0 | 67.8 | 49.3 | 83.5 | 1335.9 | 61.4 | 53.8 | 63.3 | 62.5 | 52.9 | 16.1 | 23.4 | 94.6 |
| 2  Self-Filter (*AL & CL*) | 73.7 | 58.3 | 53.2 | 61.4 | 52.9 | 83.8 | 1306.2 | 48.8 | 45.3 | 64.9 | 60.5 | 48.7 | 14.1 | 19.8 | 90.1 |
| 3  EL2N (*AL Only*) | 76.2 | 58.7 | 43.7 | 65.5 | 53.0 | 84.3 | 1439.5 | 53.2 | 47.4 | 64.9 | 61.8 | 49.3 | 16.5 | 23.9 | 93.4 |
| 4  SemDeDup | 74.2 | 54.5 | 46.9 | 65.8 | 55.5 | 84.7 | 1376.9 | 52.2 | 48.5 | 70.0 | 60.9 | 53.5 | 15.8 | 24.2 | 94.1 |
| 5  D2-Pruning | 73.0 | 58.4 | 41.9 | 69.3 | 51.8 | 85.7 | 1391.2 | 65.7 | 57.6 | 63.9 | 62.1 | 52.5 | 15.3 | 22.3 | 94.8 |
| 6  COINCIDE (*AL Only*) | 76.5 | 59.8 | 46.8 | 69.2 | 55.6 | 86.1 | 1495.6 | 63.1 | 54.5 | 67.3 | 62.3 | 53.3 | 16.1 | 24.3 | 97.8 |
| 7  Random | 75.7 | 58.9 | 44.3 | 68.5 | 55.3 | 84.7 | 1483.0 | 62.2 | 54.8 | 65.0 | 61.7 | 50.2 | 15.1 | 21.9 | 95.0 |
| 8  CLIP-Score | 73.4 | 51.4 | 43.0 | 65.0 | 54.7 | 85.3 | 1331.6 | 55.2 | 52.0 | 66.2 | 61.0 | 49.1 | 14.3 | 20.3 | 90.6 |
| 9  Perplexity (*AL Only*) | 75.8 | 57.0 | 47.8 | 65.1 | 52.8 | 82.6 | 1341.4 | 52.0 | 45.8 | 68.3 | 60.8 | 48.7 | 14.5 | 20.9 | 91.1 |
| **PROGRESS (*AL & CL*)** | | | | | | | | | | | | | | | |
| 10  Loss as Obj. | 75.7 | 58.6 | 49.6 | 70.1 | 55.1 | 86.3 | 1498.4 | 62.5 | 55.5 | 65.5 | 63.4 | 53.3 | 17.3 | 23.7 | 98.4 |
| 11  Accuracy as Obj. | 75.2 | 58.8 | 53.4 | 69.9 | 55.1 | 85.9 | 1483.2 | 61.1 | 54.4 | 65.5 | 63.0 | 52.8 | 17.3 | 24.6 | **98.8** |
| **LLaVA-v1.5-13B** | | | | | | | | | | | | | | | |
| 12  Full-Finetune | 80.0 | 63.3 | 58.9 | 71.2 | 60.2 | 86.7 | 1541.7 | 68.5 | 61.5 | 69.5 | 68.3 | 60.1 | 19.3 | 22.1 | 100 |
| 13  Self-Sup (*AL Only*) | 76.3 | 60.5 | 50.0 | 70.2 | 52.7 | 85.4 | 1463.8 | 63.7 | 57.6 | 64.9 | 65.2 | 53.3 | 17.2 | 23.2 | 93.8 |
| 14  Self-Filter (*AL & CL*) | 75.0 | 59.8 | 48.6 | 69.5 | 55.8 | 84.5 | 1446.9 | 58.8 | 51.8 | 69.1 | 65.3 | 52.4 | 16.9 | 23.1 | 92.6 |
| 15  EL2N (*AL Only*) | 77.2 | 59.6 | 54.8 | 69.9 | 56.1 | 84.1 | 1531.0 | 59.3 | 52.3 | 65.8 | 65.7 | 53.9 | 17.0 | 24.4 | 94.4 |
| 16  SemDeDup | 75.6 | 57.5 | 48.3 | 70.5 | 57.7 | 85.3 | 1397.6 | 59.0 | 51.1 | 68.7 | 64.9 | 53.2 | 16.8 | 24.6 | 92.9 |
| 17  D2-Pruning | 73.9 | 60.5 | 49.8 | 70.4 | 55.2 | 84.9 | 1463.0 | 67.3 | 59.9 | 66.5 | 65.9 | 53.4 | 16.9 | 23.5 | 94.7 |
| 18  COINCIDE (*AL Only*) | 77.3 | 59.6 | 49.6 | 69.2 | 58.0 | 87.1 | 1533.5 | 64.5 | 56.6 | 66.4 | 65.9 | 52.9 | 18.4 | 25.0 | 95.9 |
| 19  Random | 76.7 | 60.5 | 48.0 | 68.8 | 57.7 | 84.8 | 1484.9 | 62.8 | 55.2 | 68.6 | 65.5 | 57.9 | 17.1 | 24.3 | 95.0 |
| 20  CLIP-Score | 75.3 | 52.6 | 42.2 | 69.7 | 57.3 | 85.4 | 1426.3 | 60.4 | 54.0 | 68.1 | 63.3 | 52.8 | 17.4 | 23.7 | 91.8 |
| 21  Perplexity (*AL Only*) | 77.0 | 58.5 | 48.2 | 68.7 | 54.8 | 83.1 | 1508.8 | 57.5 | 50.3 | 68.7 | 64.7 | 53.1 | 17.6 | 23.8 | 92.7 |
| **PROGRESS (*AL & CL*)** | | | | | | | | | | | | | | | |
| 22  Loss as Obj. | 76.8 | 59.7 | 54.6 | 70.4 | 58.0 | 87.2 | 1458.3 | 63.8 | 56.9 | 69.9 | 65.1 | 58.0 | 17.9 | 24.6 | **96.8** |
| 23  Accuracy as Obj. | 76.9 | 58.9 | 53.0 | 70.1 | 57.5 | 87.1 | 1497.6 | 63.9 | 57.6 | 67.3 | 65.4 | 57.7 | 18.0 | 24.5 | 96.5 |

(LLaVA-Bench(Liu et al., 2023b)), fine-grained skills (MME (Liang et al., 2024), MMBench-en (Liu et al., 2024b), SEED (Li et al., 2023a)), and scientific questions and diagrams (SQA-I (Lu et al., 2022), AI2D (Kembhavi et al., 2016), ChartQA (Masry et al., 2022)).

**Evaluation Metrics.** Following all prior work, we use *standard evaluation metrics* to ensure consistency and fair comparison. Specifically, we report **average relative performance** (Lee et al., 2024) across benchmarks to provide a unified measure of generalization. For each benchmark, relative performance is defined as: $\text{Rel.} = \left( \frac{\text{Model Performance}}{\text{Full Data Finetuned Performance}} \right) \times 100\%$. This normalizes differences in performance scale and difficulty of different benchmarks and is consistent with prior work.

## 4.2  RESULTS AND ANALYSIS

**PROGRESS is more effective than existing SOTA in data efficient learning.** Table 1 (Row 0-11) compares PROGRESS against state-of-the-art baselines for training **LLaVA-v1.5-7B** on LLaVA-665K dataset under a 20% data budget, following *standard protocol*. PROGRESS achieves the highest relative performance (98.8%), outperforming all baselines, including those requiring access to ground-truth answers for the entire dataset and additional reference VLMs. In contrast, PROGRESS uses supervision only on a *need basis* for 20% of samples and relies solely on self-supervised features, yet reaching near-parity with full finetuning. Beyond aggregate gains, PROGRESS also ranks among

---

[3]Reproduced with official code.

Table 2: **Architecture and Dataset Generalization.** For Architecture Generalization, we report Qwen2-VL-7B on the LLaVA-665K dataset using 20% sampling ratio. For Dataset Generalization, we report LLaVA-v1.5-7B on Vision-Flan dataset using 16.7% sampling ratio following prior work.

| Method | VQAv2 | GQA | VizWiz | SQA-I | TextVQA | POPE | MME | MMBench en | cn | LLaVA-Wild | SEED | Rel. (%) |
|---|---|---|---|---|---|---|---|---|---|---|---|---|
| | | | | | **Architecture Generalization (Qwen2-VL-7B)** | | | | | | | |
| Full-Finetune | 77.4 | 61.7 | 45.5 | 81.4 | 59.7 | 84.3 | 1567.9 | 76.1 | 75.1 | 84.8 | 66.9 | 100 |
| Random | 76.2 | 60.1 | 43.6 | 81.4 | 58.7 | 83.7 | 1556.8 | 76.8 | 74.5 | 81.7 | 67.6 | 98.7 |
| COINCIDE | 76.7 | 60.2 | 45.4 | 81.7 | 59.4 | 83.6 | 1583.5 | 77.4 | 76.2 | 80.5 | 67.9 | 99.6 |
| **PROGRESS** | 76.2 | 60.5 | 47.1 | 82.3 | 58.0 | 84.3 | 1560.1 | 77.2 | 72.9 | 87.1 | 67.6 | **100.0** |
| | | | | | **Dataset Generalization (Vision-Flan-191K)** | | | | | | | |
| Full-Finetune | 69.4 | 46.0 | 49.7 | 59.9 | 34.1 | 85.1 | 1306.1 | 49.1 | 51.7 | 35.7 | 53.3 | 100 |
| Random | 66.0 | 43.8 | 52.2 | 62.1 | 39.7 | 82.7 | 1072.2 | 48.7 | 43.7 | 40.4 | 28.7 | 95.0 |
| COINCIDE | 66.3 | 43.6 | 51.0 | 63.8 | 35.2 | 81.9 | 1222.2 | 56.7 | 45.5 | 31.1 | 37.5 | 95.8 |
| **PROGRESS** | 65.5 | 44.0 | 53.6 | 62.5 | 42.0 | 82.9 | 1040.9 | 43.6 | 47.4 | 43.2 | 45.3 | **99.0** |

the top two methods on 8 out of 14 benchmarks, showing strong generalization across diverse tasks (Table 1 (Row 0-11))—*e.g.*, including perception-focussed VQAv2 (75.2), scientific questions and diagrams (ChartQA:17.3, AI2D:52.8), and object hallucination POPE (85.9). Notably, it exceeds full-data performance on VizWiz (53.4 vs. 47.8), SQA-I (69.9 vs. 68.4), MME (1483.2 vs. 1476.9), ChartQA (17.3 vs. 16.4) and CMMMU (24.6 vs. 22.1). These results demonstrate PROGRESS is a dynamic and fully automated alternative for efficient VLM training under limited supervision.

**Scalability to Larger Models.** To assess scalability, we use PROGRESS to train the larger **LLaVA-v1.5-13B** model under the same 20% data budget, testing whether our method developed for LLaVA-v1.5-7B transfers effectively to a higher-capacity model without hyperparameter tuning. As shown in Table 1 (Row 12-23), PROGRESS achieves a relative performance of 96.8%, outperforming all baselines. Beyond aggregate gains, PROGRESS ranks among the top-2 methods on 8 out of 14 benchmarks compared with all baselines, demonstrating strong generalization.

**Architectures and Dataset Generalization.** In Table 2, we test generalization of PROGRESS across different VLM architecture and IT dataset with accuracy as signal. For **architecture generalization**, we use newer **Qwen2-VL-7B** and train it on the LLaVA-665K dataset using the same 20% data budget and identical hyperparameters. We compare PROGRESS with two of the strongest (highest performing) established baselines—Random Sampling and COINCIDE—across multiple multimodal benchmarks. PROGRESS achieves the highest overall relative performance of 100% and ranks first or second on 9 out of 11 benchmarks (Tab. 2, **top**). We also report results on **Qwen2.5-VL-32B-Instruct** (See Appendix B.1), demonstrating the strong generalization to even larger-scale VLMs. For **dataset generalization**, we report LLaVA-v1.5-7B on Vision-Flan dataset under a stricter 16.7% annotation budget (***standard protocol***) to assess generalization in low-resource settings. PROGRESS achieves the highest overall relative performance of 99.0%, outperforming COINCIDE (95.8%) and Random (95.0%) and ranks first or second on 8 out of 11 benchmarks (Tab. 2, **bottom**). These results underscore the calability and generalization of PROGRESS , making it a practical solution for efficient training across diverse architecture and datasets.

## 4.3 INVESTIGATING THE EFFECTIVENESS OF DIFFERENT COMPONENTS OF PROGRESS

Here, we analyze & ablate the components of our method. We use LLaVA-v1.5-7B on LLaVA-665K with 20% sampling and accuracy as the default objective unless stated otherwise.

**How effective is our Selection Policy (i.e Eqn 1)?** We evaluate the efficacy of PROGRESS (selection based on relative accuracy change; Sec. 3.2) against several competitive strategies in Table 3. As a reference, we include

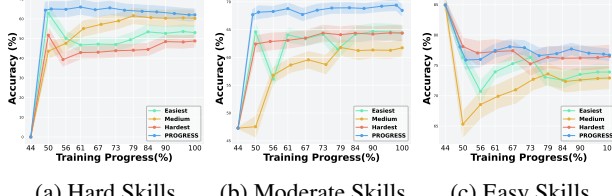

(a) Hard Skills  (b) Moderate Skills  (c) Easy Skills

Figure 4: **Learning Dynamics Across Difficulty Levels.** PROGRESS consistently achieves higher accuracy and reduced variance compared to other selection strategies

warm-up only model (row 1), a DINO–BERT variant of COINCIDE (row 3), which replaces Tiny-LLaVA features with the same unsupervised features used in our method—testing whether the

Table 3: **Ablation of Selection Policy.** Comprehensive performance comparison with baselines and curriculum learning (i.e *CL*) based selection strategies.

| Method | VQAv2 | GQA | VizWiz | SQA-I | TextVQA | POPE | MME | MMBench en | MMBench cn | LLaVA-Wild | SEED | AI2D | ChartQA | CMMMU | Rel. (%) |
|---|---|---|---|---|---|---|---|---|---|---|---|---|---|---|---|
| 0  Full-Finetune | 79.1 | 63.0 | 47.8 | 68.4 | 58.2 | 86.4 | 1476.9 | 66.1 | 58.9 | 67.9 | 67.0 | 56.4 | 16.4 | 22.1 | 100 |
| 1  Warm-up Only | 73.1 | 55.9 | 43.8 | 67.9 | 54.2 | 85.4 | 1410.3 | 58.5 | 52.7 | 64.6 | 60.5 | 52.4 | 16.1 | 24.5 | 94.6 |
| 2  Random | 75.7 | 59.0 | 43.8 | 68.8 | 54.9 | 85.6 | 1414.2 | 61.9 | 54.9 | 66.2 | 63.3 | 48.6 | 17.3 | 25.2 | 96.8 |
| 3  (DINO-BERT) COINCIDE | 76.0 | 58.3 | 40.1 | 67.8 | 55.7 | 87.2 | 1466.1 | 62.2 | 53.8 | 69.1 | 63.3 | 52.6 | 17.6 | 23.7 | 96.9 |
| 4  Easiest (*CL*) | 72.0 | 54.8 | 50.2 | 67.1 | 51.6 | 85.7 | 1407.4 | 57.0 | 52.6 | 65.2 | 59.5 | 50.1 | 12.3 | 22.8 | 92.3 |
| 5  Medium (*CL*) | 69.3 | 52.5 | 46.0 | 68.3 | 50.8 | 85.4 | 1307.6 | 54.6 | 48.7 | 62.5 | 57.7 | 47.6 | 14.3 | 26.1 | 91.1 |
| 6  Hardest (*CL*) | 72.8 | 54.8 | 52.1 | 61.3 | 50.5 | 85.4 | 1364.8 | 37.9 | 34.5 | 67.5 | 54.1 | 41.4 | 15.8 | 25.9 | 88.5 |
| **PROGRESS** | | | | | | | | | | | | | | | |
| 7  Loss as Obj. | 75.7 | 58.6 | 49.6 | 70.1 | 55.1 | 86.3 | 1498.4 | 62.5 | 55.5 | 65.5 | 63.4 | 53.3 | 17.3 | 23.7 | 98.4 |
| 8  Accuracy as Obj. | 75.2 | 58.8 | 53.4 | 69.9 | 55.1 | 85.9 | 1483.2 | 61.1 | 54.4 | 65.5 | 63.0 | 52.8 | 17.3 | 24.6 | **98.8** |

advantage of PROGRESS stems simply from the warmup/feature choice or from the progress-driven selection policy. We further compare with curriculum baselines: **Random Sampling**, **Easiest-Sampling** (selecting clusters with highest absolute accuracy at a given step), **Medium-Sampling** (selecting mid-accuracy clusters), and **Hardest-Sampling** (selecting lowest-accuracy clusters).

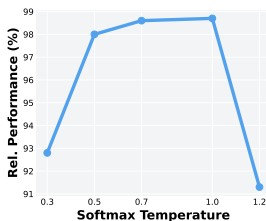

Figure 5: **Ablation of Softmax Temperature.** Both very-low and very-high temperatures lead to a significant performance drop.

As shown in Tab. 3, PROGRESS achieves the highest relative score (98.8%), ranking first on 7 out of 14 benchmarks and second on 5 others. To further analyze learning dynamics across strategies, we group skill clusters into difficulty levels - easy, moderate, and hard based on their initial accuracy, and track the average performance of 50 clusters per level (see Fig. 4). PROGRESS consistently achieves higher mean performance with lower variance across all levels, effectively balancing learning across task difficulties.

**How important is balancing informativeness & diversity in selected samples (i.e Eqn 2)?** We ablate the *temperature* $\tau$ in the softmax skill-selection (Eqn. 2). As discussed in Sec 3.2, lower $\tau$ would over-focuses on top clusters, reducing diversity, while very high $\tau$ makes high-improvement clusters lose priority. Fig. 5 shows, $\tau = 1.0$ achieves the best performance (98.8% Rel.), balancing priority and diversity. Decreasing $\tau$ (0.7, 0.5, 0.3) degrades performance, with lowest $\tau = 0.3$ dropping to 92.8% (-6%), confirming that overly sharp distributions cause *mode collapse*. Excessive diversity ($\tau = 1.2$) also hurts performance, as high-improvement clusters lose their clear priority.

**Wall Clock Comparison?** Consistent with prior work (Lee et al., 2024), we measure the wall-clock cost of the *entire pipeline*—data selection plus model finetuning—against relative performance (Rel.). As shown in Fig. 6, PROGRESS reaches relative performances of 96.3%, 97.1%, 98.2%, 99.1%, and 100% within 1.5, 1.75, 2.5, 4, and 5.67 hours, respectively, making it Pareto-superior to COINCIDE. Furthermore, Full-data finetuning requires ~9 hours (and needs 100% data for training), much higher than our method which needs only 5.67 hr of total training time and uses only 20% samples for training. **Note-** for fair comparison (Fig 6), runtime includes all cost for entire pipeline, specifically: PROGRESS comprises feature extraction, K-means clustering, self-evaluation, and training, while COINCIDE includes feature extraction, clustering, and training. Both methods use same GPU

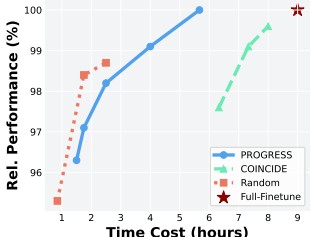

Figure 6: **Wall-clock time** comparison. We show results on LLaVA-665K here and Vision-Flan-191K in Appendix B.3.

compute and 20% data to train following standard protocol from prior work. See Appendix B.4 for more detailed time breakdown for each stage & Appendix B.5 for reduced annotation cost analysis.

**How effective is PROGRESS under different data budget?** Fig 7 shows performance under different data budget used for training. PROGRESS consistently outperforms strongest baselines - Random and COINCIDE across different sampling ratios, highlighting its effectiveness. Notably, on scaling data size to higher percentages 32%, 64%, our method outperforms full-data-finetuning (which uses 100% data) by larger and larger margin. See Appendix B.6 for more details on scaling.

**Ablation on Hyperparameters** (K,$\gamma$, warm-up ratio, etc) - See Appendix B.2.

**Importance of order of skill acquisition?** See Appendix B.2.

### 4.4 ANALYZING MODEL LEARNING BEHAVIOR

**How does benchmark difficulty and data frequency impact performance?** Here, we further examine *how improvements vary with benchmark difficulty and data frequency*. In Fig. 8(a), we plot accuracy gains over the warm-up-only model as a function of difficulty, defined as the gap from full-data finetuning $((100 - \text{full-finetune score})/100)$ (details in Appendix B.7.1). PROGRESS achieves the largest gains on benchmarks of moderate difficulty. At the extremes, easy tasks such as POPE show limited improvements due to saturation, while hard tasks such as ChartQA, CMMMU yield smaller gains because chart-related skills & rare Chinese-language skills are underrepresented in LLaVA-665K dataset($\sim$0.96% & $\sim$1.1% respectively).

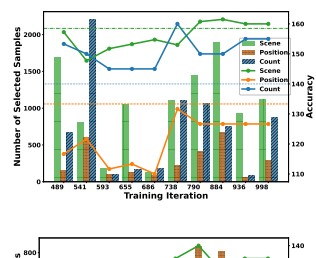

Figure 7: **Ablation with different data budget for training**. Relative performance on Vision-Flan dataset under different sampling ratio.

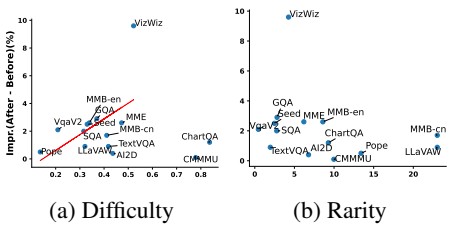

(a) Difficulty      (b) Rarity

Figure 8: **Accuracy improvements** with **(a) benchmark difficulty** and **(b) sample rarity**. Largest performance gains occur in mid-range of difficulty & low-mid range of rarity.

Despite this scarcity, PROGRESS still surpasses full-data finetuning on both ChartQA and CMMMU with only 20% training data (Table 1, rows 10–11), showing that it can *enhance* rare or niche abilities under limited supervision, though scarcity naturally constrains absolute gains. Fig. 8(b) further confirms this trend by plotting improvements against rarity, measured as $\log(1/\text{frequency})$ of benchmark-aligned samples (details in Appendix B.7.2): once again, PROGRESS performs best in the mid-rarity regime, where skills are neither over-abundant (with limited additional benefit from more samples) nor rare (too few samples to generalize). Our findings align with the Zone of Proximal Development (Vygotsky, 2012) & findings in (Khan et al., 2025), which state that learning is most effective just beyond current ability—neither too easy nor too difficult.

**What skills are prioritized during training?** We analyze which skills the model acquires at different stages of training using fine-grained skill categories from prior work (Liang et al., 2024), which provide interpretable skills. Results in Fig. 9 show (a) *grounded perceptual skills* (scene, position, count) and (b) *language/symbolic skills* (OCR, text translation, code reasoning). Each cluster is assigned a dominant ability (see Appendix B.8), & we track number of samples selected per ability (bars) along with accuracy throughout training. We find that some abilities consistently outperform others (e.g., scene > position, OCR > translation/code). Early in training (iteration 541), the model prioritizes *count*, but accuracy does not improve until it shifts to easier abilities first (e.g., scene) and later revisits *count* and *position* (iteration 738), where accuracy now increases and stabilizes. For language skills, OCR initially declines but the model gradually selects more OCR samples & improves, peaking at iteration 738 with largest gain. Toward the end (after iteration 790), the model increasingly focuses on even harder abilities such as *code reasoning* and *text translation*, with both sample selection and performance rising. Notably, PROGRESS surpasses full-data finetuning (dashed line) on these challenging skills despite using only 20% data.

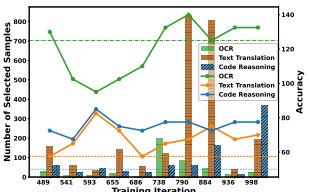

Figure 9: **What skills model prioritize and when?** We track the number of samples selected per ability (bars) and corresponding accuracy trends during training, with dashed lines indicating full-data finetuning performance.

### 5 CONCLUSION

We introduce PROGRESS, a dynamic and data-efficient framework for instruction-tuning VLMs under a strict data and supervision budget. By tracking learning progress across unsupervised skill clusters and prioritizing samples that are most learnable at each stage, PROGRESS effectively controls both the acquisition and order of skills. Our method achieves near full-data performance with just 16–20% supervision while requiring no additional reference VLMs, requires annotations only on *need basis* , and scales across architectures and datasets. Extensive experiments show that this self-paced, progress-driven strategy outperforms strong baselines while offering practical advantages in scalability, diversity, and data efficiency.

## REPRODUCIBILITY STATEMENT

To ensure reproducibility of our results, we provide comprehensive implementation details in Section 4.1 and Appendix A.2. Our experimental setup follows standard protocols established in prior data-efficient learning works (Lee et al., 2024; Wu et al., 2025; Chen et al., 2024a). All datasets used are publicly available: Vision-Flan-191K (Xu et al., 2024) and LLaVA-665K (Liu et al., 2023b). Model architectures include LLaVA-v1.5-7B, LLaVA-v1.5-13B (Liu et al., 2023b), Qwen2-VL-7B (Wang et al., 2024), and Qwen2.5-VL-32B-Instruct (Bai et al., 2025), all of which are open-sourced. We use official training codebases: the original LLaVA repository for LoRA training LLaVA models, official Qwen code for full-parameter training Qwen2-VL-7B, and LLaMA-Factory for LoRA training Qwen2.5-VL-32B-Instruct. All reported metrics are averaged over 3 independent runs with different random seeds to ensure statistical reliability. The complete data selection pipeline, including feature extraction details, clustering procedures, and accuracy evaluation prompts, is described in Section 3 and Appendix A. Our code implementation, including data selection algorithms, training scripts, and evaluation pipelines, will be **made publicly available** upon publication to facilitate reproduction and future research.

## ETHICS STATEMENT

This work improves data- and compute-efficient training of vision–language models using only publicly available datasets, adhering to their licenses. Our method reduces annotation and compute costs, lowering the environmental and financial burden of large-scale training. No new human data were collected, and no personally identifiable or sensitive information is involved. As with prior work, models may inherit biases from the underlying datasets; we encourage future efforts toward more diverse and inclusive data. Broader risks of VLM misuse (e.g., misinformation) remain, though efficiency improvements make research more accessible and sustainable.

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

# Appendix

In this appendix section, we provide additional details that could not be included in the main paper due to space constraints:

- Additional details on the **Baselines** used, including setup and implementation (extending Sec.4.1 and Fig. 1 of the main manuscript).
- **Implementation** details and **hyperparameter** settings (extending Sec.4.1).
- Comparison with concurrent work ICONS (extending Sec.4.2 and Sec 2).
- **Intuition and Justification** for using relative improvement (extending Sec.3).
- Further Ablation studies on the **Scalability and Fine-tuning, Effect of Hyperparameters, Wall-clock Time Analysis, Time Breakdown Analysis, Annotation Time Analysis** and **Scaling Performance** (extending Sec.4.3).
- **Word Cloud Visualization** for our multimodal clustering approach (extending Sec.3.1).
- Visualization of the **Diversity** of samples selected by our method (extending Sec. 4.3).
- Additional details on the setup and algorithms used for our analysis (extending Sec.4.4).
- Limitations of our method and LLM usage disclosure statement

## A   DETAILS OF EXPERIMENTAL SETUPS

### A.1   BASELINES

We follow the standard experimental settings and implementation protocols for all baselines as established in recent prior work (see COINCIDE (Lee et al., 2024)), using official code. For completeness, clarity and reproducibility, we additionally provide detailed descriptions of each baseline here.

**COINCIDE** (Lee et al., 2024) is a strong coreset selection method that leverages concept-based clustering and mutual transferability between clusters to guide sample selection. It performs coreset selection only once before training by clustering internal activations of a separately trained additional reference VLM (e.g., TinyLLaVA(Zhou et al., 2024)). It relies on static selection strategy that does not adapt to the model's learning progress, requires an additional pretrained VLM, ground-truth annotations for full dataset to extract activation maps, and manual intervention to select appropriate activation layers—making the method resource-intensive and difficult to scale.

**CLIP-Score** (Hessel et al., 2022) It ranks image-instruction pairs based on visual-textual similarity computed by the CLIP model (Radford et al., 2021), selecting top-scoring samples for training. While this approach assumes that higher similarity indicates greater informativeness, it relies on static, precomputed metrics that do not adapt to the model's learning progress. Prior work (Lee et al., 2024) has shown that such metrics often fail to capture important data modes, resulting in reduced diversity and suboptimal generalization—limitations that PROGRESS overcomes through dynamic, progress-driven selection.

**EL2N** (Paul et al., 2021) ranks training samples based on the expected L2-norm of prediction error: $\mathbb{E}[\|p(x) - y\|_2]$ where $p(x)$ is the token distribution predicted by a reference VLM, $x$ is the input, and $y$ is the ground-truth label. This score reflects how confidently and accurately the reference model predicts each sample. However, it requires access to a fully trained additional VLM and ground-truth labels for the entire dataset, making it resource-intensive and static.

**Perplexity** (Marion et al., 2023) measures the uncertainty in the model's predictions and is defined as $\exp(-\mathbb{E}[\log p(x)])$, where $p(x)$ denotes the likelihood assigned to input $x$ by a additional reference VLM model. Samples from the middle of the perplexity distribution are selected, following prior work (Lee et al., 2024). However, it requires access to a fully trained additional VLM and, like other static metrics, often fails to capture important data modes—potentially limiting diversity and downstream generalization.

**SemDeDup** (Abbas et al., 2023) aims to reduce redundancy by removing semantically duplicated samples. It clusters the output embeddings of the final token from a reference model's last layer and

retains a diverse subset by eliminating near-duplicates and reducing redundancy. This method also requires an additional reference VLM to extract the final token features and ground-truth labels for the entire dataset.

**D2-Pruning** (Maharana et al., 2024) constructs a graph over training data where nodes encode sample difficulty and edges capture pairwise similarity. It selects a diverse coreset by pruning this graph while preserving representative and challenging samples. Difficulty is measured using the AUM score, defined as $p_y(x) - \max_{i \neq y} p_i(x)$, where $p_y(x)$ is the model's confidence for the ground-truth label $y$. Similarity is computed using the L2 distance between average final-layer token embeddings from a additional reference VLM. This method requires access to an additional reference VLM for embedding extraction and scoring and ground-truth labels for the entire dataset.

**Self-Sup** (Sorscher et al., 2022) clusters data using averaged output embeddings from the final-layer tokens of a reference model. It assigns scores based on distance to cluster centroids, selecting samples that are closest—assumed to be the most prototypical representatives of the data distribution. This method also requires access to an additional reference VLM for embedding extraction.

**Self-Filter** (Chen et al., 2024a) is a recent coreset selection method originally proposed for the LLaVA-158k dataset (containing three vision-language tasks). It jointly fine-tunes a scoring network alongside the target VLM on the *entire labeled dataset*, using this as learned reference model to score and filter training samples—hence it requires an additional reference model trained on full data with full annotations. Following previous work (Lee et al., 2024), we adopt the stronger variant that also incorporates CLIP scores and features.

**Random.** We additionally report results for *Random*, which finetunes the model using a coreset selected via random sampling. Despite its simplicity, Random serves as a strong and competitive baseline—prior work (Lee et al., 2024)has shown that random sampling often preserves sample diversity and can outperform more complex selection methods in certain settings.

*Note:* We use standard setup for the baseline implementations as described in prior work (see COINCIDE Appendix (Lee et al., 2024)). For **COINCIDE**, **EL2N**, **SemDeDup**, **D2-Pruning**, and **Self-Sup**, we use image, question, and ground-truth answer for full dataset as inputs along with additional reference VLM (i.e TinyLLaVA) to extract features following prior work (Lee et al., 2024). **Self-Filter** requires full dataset to finetune additional reference network—the score-net. As a result, these baselines require an additional reference vision-language model or full dataset annotations (100%) or both.

**Active Learning Nature of the Compared Baselines** AL selects which unlabeled samples to label next using acquisition function a(x) (Weng, 2022). Higher a(x) (e.g., uncertainty, error, diversity, representativeness) implies higher expected utility of sample. EL2N (Paul et al., 2021) and Perplexity (Marion et al., 2023) are uncertainty-based AL methods with error/uncertainty as acquisition functions. Self-Filter (Chen et al., 2024a) uses deep network as acquisition function. It jointly trains a deep score network with the target model to later rank and select samples. COINCIDE (Lee et al., 2024) is hybrid sampling based AL which uses a hybrid of skill transferability + diversity for acquisition of samples to select a representative coreset. Self-Sup (Sorscher et al., 2022) aligns with representativeness-based AL coreset selection, aiming to pick prototypical samples that best cover the data distribution.

**Curriculum Learning Baselines** Curriculum Learning (CL) methods are majorly two types:

Type 1) External model decides difficulty and select samples.
Type 2) Models own feedback decides difficulty and select samples.

Self-Filter (Chen et al., 2024a) is an instantiation of Type 1 CL methods where an external model (i.e deep score network) is jointly trained with the target model to quantify the learning difficulty of sample and select hard-examples for training. The auxiliary score network guides selection of samples expected to have significant value for model training based on difficulty

We also compare with Type 2 curriculum policies in Table 3: Easiest Selection (Table 3 row 4) - model self-evaluates and chooses easy-samples with highest absolute performance, Medium difficulty Selection (Table 3, row 5)- model self-evaluates and chooses samples with medium absolute performance aligning with zone-of-proximal development, Hardest Selection (Table 3, row 6) - model

self-evaluates and chooses hard-samples with lowest absolute performance Relative-Improvement based selection (ours, Table 3, row 7-8)

## A.2 IMPLEMENTATION DETAILS.

In this section, we provide elaborate details on implementation of our approach in continuation to the brief details we provide in Section 4.1 of main manuscript.

We first partition the unlabeled data pool $\mathbb{U}$ into $K$ skill clusters using spherical k-means, following the fully *unsupervised* concept categorization procedure described in Section 3.1. Training begins with a brief warmup phase (see details in Appendix A.3), which equips the model with basic instruction-following capability and ensures that skill-level performance estimates are reliable in the beginning of training.

Subsequently, we apply our Prioritized Concept Learning (PCL) strategy (see Section 3.2) to estimate the expected performance improvement for each skill cluster between iteration

Table 4: Hyperparameter configurations. $K$ represents the number of clusters.

| Method | LLaVA-1.5 | Vision-Flan |
|---|---|---|
| CLIP-Score | high score selected | high score selected |
| EL2N | medium score selected | medium score selected |
| Perplexity | medium score selected | medium score selected |
| SemDeDup | $K$ : 10,000 | $K$ : 5,000 |
| D2-Pruning | $k$ : 5, $\gamma_r$ : 0.4, $\gamma_f$ : 1.0 | $k$ : 5, $\gamma_r$ : 0.4, $\gamma_f$ : 1.0 |
| Self-Sup | $K$ : 10,000 | $K$ : 5,000 |
| Self-Filter | $k$ : 10, $\gamma$ : 1 | $k$ : 10, $\gamma$ : 1 |
| COINCIDE | $K$ : 10,000, $\tau$ : 0.1 | $K$ : 5,000, $\tau$ : 0.1 |
| **PROGRESS** | | |
| **Warmup Stage** | | |
| Number of Clusters | $K$ : 10,000 | $K$ : 5,000 |
| Warmup Ratio (w.r.t full data) | 9% | 8.4% |
| **Prioritized Concept Learning** | | |
| Number of Clusters | $K$ : 1,000 | $K$ : 200 |
| Temperature of Softmax | $\tau$ : 1.0 | $\tau$ : 1.0 |
| BatchSize | 128 | 128 |
| Selection Gap | $\gamma * BatchSize$ : 7,500 | $\gamma * BatchSize$ : 3,500 |
| Random Exploration | $\delta\%$ : 10 % | $\delta\%$ : 10 % |

$t$ and $t - \gamma$ (as defined in Eqn 1), using either accuracy or loss as the tracking metric (see Table 1, row 10 and 11). For the accuracy-based variant, we compute cluster-wise accuracy using an LLM judge as metric that compares the VLM output to ground-truth answers—though this is not required for our loss-based variant. Samples are then selected using a temperature-controlled softmax over the improvement scores (see Eqn 2). This selection process is repeated every $\gamma$ iterations, and in each round, we sample a total of $\gamma * BatchSize$ examples for annotation and training. We refer to this $\gamma * BatchSize$ as selection gap from here on.

**Hyperparameters for Baselines and PROGRESS** To ensure fair comparison, we use the same hyperparameters as COINCIDE (Lee et al., 2024) for all baselines. The hyperparameters for both the baselines and PROGRESS are summarized in Table 4. For model training, we apply LoRA (Hu et al., 2021) to LLaVA-v1.5 and follow the official fine-tuning settings provided in the LLaVA-1.5 release. For Qwen2-VL, we perform full fine-tuning using the official hyperparameters specified by Qwen2-VL. For Qwen2.5-32B-Instruct, we apply LoRA and follow the official fine-tuning settings provided in LLaMA-Factory (Zheng et al., 2024). For accuracy estimation, we use LLMs such as InternLM2-Chat-20B (Cai & et al., 2024) as the judge. We provide the question, ground-truth answer, and predicted answer (without the image) as input and ask the LLM to decide whether the prediction is correct. The full prompt is shown below. Ablation studies on all hyperparameters are provided in Section 4.3 and Appendix B.

## A.3 WARMUP PHASE

Following prior work (Xia et al., 2024; Wu et al., 2025), we begin with a brief warmup phase using a small subset—9% of the total dataset pool size (see ablation in Fig. 10 (a))—to equip the model with basic instruction-following capability and ensure that skill-level performance estimates are credible at the start of training—when the model is still untrained. These early estimates are crucial for tracking relative learning progress across skills in subsequent training phases.

To be effective, the warmup set should provide broad skill coverage across diverse clusters and include transferable examples that support generalization to unseen skills that model will later learn. To this end, we adopt the sampling strategy from Lee et al. (2024), selecting samples from each

---

**Prompt for Accuracy Estimation**

Given an input question and two answers: a candidate answer and a reference answer, determine if the candidate answer is correct or incorrect.
**Rules:**

- The candidate answer is correct if it is semantically equivalent to the reference answer, even if they are phrased differently.
- The candidate answer should be marked as incorrect if it:
    - Contains factual errors compared to the reference answer
    - Only partially answers the question
    - Includes hedging language (e.g., "probably", "likely", "I think", etc.)
    - Answers a different question than what was asked
- Give a reason for your prediction.

**Output Format:**

- Answer - correct or incorrect
- Reason -

---

cluster proportionally to: $P_i \propto \exp\left(S_i/\tau D_i\right)$ where $S_i$ denotes the cluster's *transferability* and $D_i$ its *density*. This prioritizes clusters that are both diverse and likely to generalize well, ensuring a representative warmup set.

Importantly, the warmup set selection used for our method utilizes clusters generated using concatenated DINO and BERT derived features (as explained in Sec. 3.1)—instead of requiring additional reference VLMs features or ground-truth answers. As shown in Table 3 (row 1), we validate that training solely on this warmup set (without our Prioritized concept learning module in Section 3.2) yields significantly worse performance compared to our proposed strategy—highlighting the importance of our progress-driven sampling.

### A.4 COMPARISON WITH ICONS

**ICONS** (Wu et al., 2025) is a concurrent unpublished work that differs significantly from our approach. It requires **(1)** high memory and compute resources—reportedly over 100 GPU hours—to compute and store gradient-based influence scores for selection, and **(2)** access to explicit knowledge of the target task or its distribution in the form of labeled samples from validation set of target benchmarks. This assumption is impractical in general-purpose VLM training, where target tasks may be unknown at training time and usage of such high-compute refutes the goal of efficient learning. As such, ICONS is **not directly comparable** and falls outside the scope of our setting, which avoids both gradient-based selection and downstream task knowledge from target benchmarks prior to training the VLM model.

Nevertheless, we strive to compare with them in good faith by reproducing ICONS using their official codebase for fair comparison. Although ICONS has released its codebase, the validation data (for each target benchmark) it uses to simulate target task knowledge is not publicly available. The paper reports the number of validation samples used per benchmark (see Table 5), however the specific validation samples remain unspecified and are not publicly released. To approximate their setup, we randomly select an equal number of samples from the publicly available validation sets of target benchmarks and reproduce their performance.

Table 6 presents results across three settings: Row 1 shows the original ICONS results as reported; Row 2 presents our reproduction using their codebase and randomly selected validation samples; Rows 3 and 4 report results for PROGRESS . We observe that PROGRESS outperforms our ICONS reproduction in relative performance even though our method does not rely on compute-intensive gradient-based selection and does not assume any knowledge from target benchmarks, reinforcing the practicality and effectiveness of our method under realistic constraints.

Table 5: Statistics of ICONS target validation sets.

| Dataset | MME | POPE | SQA-I | MMB-en | MMB-cn | VQAv2 | GQA | VizWiz | TextVQA | LLaVA-W |
|---|---|---|---|---|---|---|---|---|---|---|
| $|\mathcal{D}_{\text{val}}|$ | 986 | 500 | 424 | 1,164 | 1,164 | 1,000 | 398 | 8,000 | 84 | 84 |
| $|\mathcal{D}_{\text{test}}|$ | 2,374 | 8,910 | 4,241 | 1,784 | 1,784 | 36,807 | 12,578 | 8,000 | 5,000 | 84 |

Table 6: Comparison between PROGRESS and ICONS. Repro. means reproductions of ICONS.

| Method | VQAv2 | GQA | VizWiz | SQA-I | TextVQA | POPE | MME | MMBench en | MMBench cn | LLaVA-Wild | SEED | AI2D | ChartQA | CMMMU | Rel. (%) |
|---|---|---|---|---|---|---|---|---|---|---|---|---|---|---|---|
| 0 Full-Finetune | 79.1 | 63.0 | 47.8 | 68.4 | 58.2 | 86.4 | 1476.9 | 66.1 | 58.9 | 67.9 | 67.0 | 56.4 | 16.4 | 22.1 | 100 |
| 1 ICONS | 76.3 | 60.7 | 50.1 | 70.8 | 55.6 | 87.5 | 1485.7 | 63.1 | 55.8 | 66.1 | - | - | - | - | - |
| 2 ICONS (Repro.) | 75.0 | 57.7 | 45.9 | 63.7 | 55.1 | 86.0 | 1434.0 | 47.1 | 37.3 | 68.4 | 57.3 | 45.3 | 17.2 | 24.3 | 91.6 |
| **PROGRESS** | | | | | | | | | | | | | | | |
| 3 Loss as Obj. | 75.7 | 58.6 | 49.6 | 70.1 | 55.1 | 86.3 | 1498.4 | 62.5 | 55.5 | 65.5 | 63.4 | 53.3 | 17.3 | 23.7 | 98.4 |
| 4 Accuracy as Obj. | 75.2 | 58.8 | 53.4 | 69.9 | 55.1 | 85.9 | 1483.2 | 61.1 | 54.4 | 65.5 | 63.0 | 52.8 | 17.3 | 24.6 | 98.8 |

## A.5 Intuition, justification, and grounding for using relative improvement and softmax-based sampling

**Empirical Validation:** We show ablation results and discussion in Table 3 for different selection policy including Our Relative improvement strategy (Table 3, row 6,7) vs Easy, Medium and Hard selection based on hard-thresholding on absolute performance. Overall, our relative improvement policy performs much better than other variants

**Intuition, justification for using relative improvement:** Prior curriculum learning work (Misra et al., 2017; Sachan & Xing, 2016; Bengio et al., 2009) research supports using relative improvement over absolute gains because it scales progress by a skill's baseline performance i.e normalizes across tasks of different difficulty and scale: tasks that are too easy (high baseline) or too hard (low baseline) tend to have small relative improvements, whereas tasks at a "moderate" difficulty yield the largest relative improvements and therefore get prioritized. This aligns with the zone-of-proximal-development (Vygotsky, 2012; Khan et al., 2025), problems that are neither too easy nor too hard produce the largest advantage, focusing on learning where progress is most productive.

In contrast, using absolute improvement would bias selection towards tasks with high raw score over-favoring raw score jumps and fail to account for the fact that a one-point increase on a hard skill is more meaningful than the same gain on an easy one, fixed thresholds likewise cannot adapt to varying skill difficulties. Overall, relative improvement avoids over-favoring weak skills with noisy raw swings and instead emphasizes areas where progress per unit of remaining error is highest, providing a stable, self-paced curriculum (Misra et al., 2017; Sachan & Xing, 2016).

**Intuition for Softmax Sampling:** We map $\Delta_k^t$ to sampling weights using a temperature-controlled softmax, $p_k \propto \exp(\Delta_k^t/\tau)$, which provides a smooth, entropy-regularized trade-off between exploiting high-utility skills and exploring others. A small floor probability safeguards rare skills. This results in a stable, noise-robust scheduler that outperforms absolute gains and hard thresholds. The temperature parameter $\tau$ explicitly balances informativeness and diversity which is crucial for effective learning and avoid mode collapse (see Fig 5).

## B Further Ablation Studies and Analysis

### B.1 Scalability and Generalization to Larger VLMs

Here we additionally present results for Qwen2.5-VL-32B-Instruct in Table 7, which we instruction-tuned using our method on the LLaVA-665K dataset, using a 20% sampling ratio following standard protocol. PROGRESS (trained with just 20% data) performs better than full- data finetuning, achieving 100.2% relative performance compared to full-data fine-tuning, demonstrating our method's scalability potential and its generalization to new large scale architectures.

Table 7: **Scalability and Fine-tuning.** We report results for Qwen2.5-VL-32B on the LLaVA-665K dataset using 20% sampling ratio.

| Method | VQAv2 | GQA | VizWiz | SQA-I | TextVQA | POPE | MME | MMBench en | MMBench cn | LLaVA-Wild | SEED | Rel. (%) |
|---|---|---|---|---|---|---|---|---|---|---|---|---|
| Full-Finetune | 83.8 | 64.5 | 58.8 | 93.1 | 82.3 | 88.5 | 1678.5 | 85.3 | 85.3 | 78.9 | 75.8 | 100 |
| **PROGRESS** | 83.7 | 63.7 | 59.0 | 93.1 | 82.0 | 88.1 | 1684.8 | 86.7 | 87.1 | 77.8 | 76.4 | **100.2** |

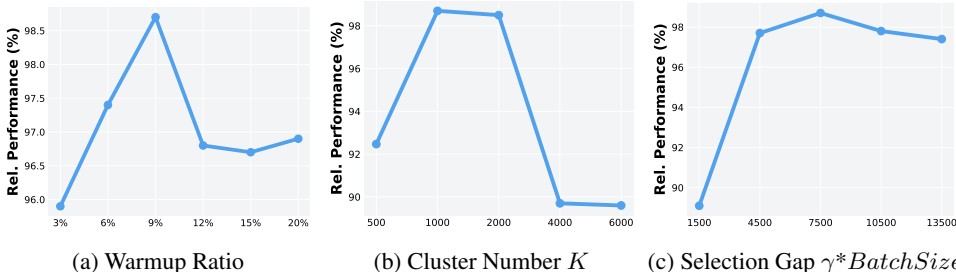

| (a) Warmup Ratio | (b) Cluster Number $K$ | (c) Selection Gap $\gamma * BatchSize$ |
|---|---|---|

Figure 10: **Ablation Studies.** (a) Effect of the warm-up ratio. (b) Effect of the number of clusters. (c) Effect of the selection gap.

## B.2 ABLATION STUDIES

**Ablations on Hyperparameter** We conduct further ablations studies in Fig. 10 on effect of hyperparameters. All experiments use LLaVA-v1.5-7B on the LLaVA-665K dataset with 20% sampling ratio and accuracy as the objective. Fig. 10(a) shows the effect of different warm-up ratios relative to the total training data pool size. Our results show that a 9% warm-up ratio achieves the best performance, as it strikes a balance between preparing the model adequately and leaving enough room for our iterative Prioritized concept learning strategy to select informative samples. The 20% warm-up ratio (i.e using only warmup selected samples to entirely train the model eleminating our Prioritized concept learning strategy completely), results in significantly reduced performance in overall relative score. Next, Fig. 10(b) shows the effect of varying the number of clusters $K$ used for our concept categorization module (Section 3.1). Using too few skill-clusters reduces skill diversity and leads to lower purity (in terms of skill types) within given cluster, while too many clusters result in redundant clusters of the same skill category and insufficient samples per cluster to yield credible accuracy estimates. We find that using approximately 1,000-2000 clusters strikes the best balance and yields optimal performance. Finally, Fig. 10(c) shows the influence of the selection gap i.e $\gamma$ *$BatchSize$ (see definition in Appendix A.2). We find that the model is particularly sensitive to small gaps; for instance, a gap size of 1,500 leads to a rapid performance decrease. Smaller gaps cause the model to switch too soon, not allowing it to learn the selected concepts sufficiently.

**How important is the order of skill acquisition?** Unlike prior methods that focus solely on selecting which samples to use (Lee et al., 2024; Wu et al., 2025), PROGRESS also controls when to introduce them during training (Section 3.2)—effectively guiding both skill selection and the order of acquisition. To assess the importance of learning order, we ablate this component by randomly shuffling the data selected by PROGRESS and training the model without respecting the intended sequence. Even with the same data, training in a random order leads to a noticeable performance drop—from 98.8% to 94.6%—highlighting that when to introduce concepts is just as important as what to learn.

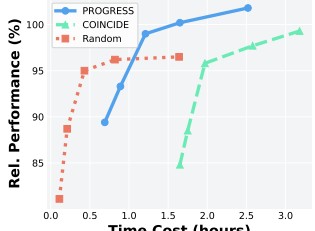

Figure 11: **Wall-clock time comparison** on Vision-Flan dataset.

## B.3 WALL-CLOCK TIME COMPARISON

We compare average relative performance (Rel.) against wall-clock time on the Vision-Flan dataset in Fig. 11. PROGRESS

(Accuracy as Objective) achieves relative performances of 89.4%, 93.3%, 99.0%, 100.2% and 101.8% with wall-clock times of 0.69, 0.89, 1.21, 1.65 and 2.52 hours. Full fine-tuning on entire dataset takes about 2.56 hours. Remarkably, our method exceeds 100% relative performance (i.e surpasses vanilla full data fine-tuning) in just 1.65 hours—including both data selection and model training time—needing only 64% of the time required for full dataset fine-tuning.

## B.4 STAGE-WISE TIME BREAKDOWN ANALYSIS

Table 8 presents a detailed time breakdown comparing our PROGRESS method with the closest competitor COINCIDE. We use same GPU compute for both COINCIDE and our method for fair comparison. COINCIDE requires over 8 hours of total computation time (plus additional unknown manual inspection time) and needs an additional pre-trained TinyLLaVA VLM to extract and store features from multiple MSA layers (4th, 8th, 12th, 16th, 20th). For each sample, they need $4096 \times 5 = 20,480$ dimensional features, which demand substantial memory resources and significantly increases clustering time. In contrast, our PROGRESS method completes in only 5.67 hours without requiring any additional VLM models. By using lightweight uni-modal self-supervised feature extractors—combining DINO-v2 (1,024 dimensions) and Sentence-BERT (384 dimensions)—we achieve efficient clustering with only 1,408-dimensional features per sample. This represents a 14.5× reduction in feature dimensionality compared to COINCIDE, while still achieving better relative performance w.r.t COINCIDE using only 20% of the training data.

Table 8: **Time breakdown comparison:** COINCIDE vs PROGRESS vs Full Data Training. We use official code to produce this time analysis

| Stage | Time | Note |
|---|---|---|
| ***Full Data Training*** | **540 min (9h)** | Training on 100% data on 4×A100 |
| ***COINCIDE*** | | |
| Manual Inspection | Unknown | Manual MSA layer selection to extract features, needs ground-truth annotations |
| Feature Extraction | 280 mins | TinyLLaVA feature extraction on 2×A100 |
| Clustering | 50 mins | With faiss-gpu KNN on single A100. COINCIDE requires large memory to store 20,480 dim. features from multiple MSA layers. |
| Training | 150 mins | Training on 20% data on 4×A100 |
| **Total** | **480 min (8h)** | |
| ***PROGRESS*** | | |
| Feature Extraction | 30 mins | Sentence-BERT (all-MiniLM-L6-v2) + DINO-v2-base on single A100 |
| Clustering | 30 mins | With faiss-gpu KNN on single A100 |
| Annotation Decision | 130 mins | Model self-evaluates and decides what to annotate next. 13 mins × 10 rounds |
| Training | 150 mins | Training on 20% data on 4×A100 |
| **Total** | **340 min (5.67h)** | |

## B.5 ANNOTATION TIME ANALYSIS: SIGNIFICANCE OF ANNOTATION COST AS THE PRIMARY BOTTLENECK

In this section, we estimate the overall annotation cost and show that it is the dominant bottleneck in scaling VLM training to larger datasets—a cost that our method drastically reduces by 80% (as it requirs only 20% data for training).

The LLaVA-665K dataset has 0.67 million samples comprising Human-curated data (OKVQA, A-OKVQA, OCRVQA, TextCaps) and Synthetic QA which still need human-annotated bounding box & object names (from COCO etc, see details in (Liu et al., 2023b; 2024a)) which are then provided to an LLM to generate Q/A.

---

**Algorithm 1** Sample Rarity Estimation via Gaussian Modeling

---

**Require:** Training dataset $\mathbb{U} = \{x_1, \ldots, x_N\}$ where $x_i = (I_i, Q_i)$; set of benchmarks $\mathcal{B} = \{B_1, \ldots, B_M\}$ with $M$ different benchmarks.

1: **Feature Extraction:** Extract DINO features from images and BERT features from questions; concatenate to form joint feature vectors, for all training and benchmark samples.
   We denote DINO-BERT embedding of $x_i$ as $\phi(x_i)$
2: **Fit Gaussian Models:** Fit multivariate Gaussian $\mathcal{N}(\mu_j, \Sigma_j)$ for each benchmark $B_j$ using its feature vectors
3: $n_j \leftarrow 0, \ \forall j \in \{1, \ldots, M\}$ {Initialize match counts for each benchmark}
4: **for** each training sample $x_i \in \mathbb{U}$ **do**
5: $\quad \ell_j = \log \mathcal{N}(\phi(x_i) \mid \mu_j, \Sigma_j), \ \forall j \in \{1, \ldots, M\}$ {Log-likelihoods under each benchmark's Gaussian}
6: $\quad k = \arg\max_j \ell_j$ {Assign to benchmark with highest likelihood}
7: $\quad n_k \leftarrow n_k + 1$ {Increment matched sample count for $B_k$}
8: **end for**
9: **for** each benchmark $B_j \in \mathcal{B}$ **do**
10: $\quad f_j = n_j/N$ {Compute frequency}
11: $\quad r_j = \log(1/f_j)$ {Compute rarity}
12: **end for**
13: **return** $\{r_1, \ldots, r_M\}$ {Rarity scores for all benchmarks}

---

Prior studies (CloudResearch, 2020) estimate it typically takes 10.3 sec for a human (Mturk worker) to annotated 1 sample (which consists Image & 4-5 Q/A pairs). So, to annotate 0.67 million samples in LLaVA dataset it will take 1902 hours (0.67 M * 10.3 / 3600).

**Estimated annotation time:**
• Full-data training (100% of 0.67M samples): ~1902 hr
• PROGRESS (20% of 0.67M samples): ~380 hr

**Overall time (annotation + training):**
• Full-data finetuning: 1902 hr (annotation) + 9 hr (training)
• PROGRESS : 380 hr (annotation) + 5.7 hr (training)

**Conclusion.** Annotation is by far the dominant cost and major bottleneck in VLM training, especially when we scale to even larger datasets. By reducing annotation to only 20%, PROGRESS drastically cuts the primary bottleneck and offers a scalable path for training on even larger datasets.

Table 9: Scaling performance of PROGRESS .

| Data Used | Rel. Score (%) |
|---|---|
| **Full Finetune** | |
| 100% | 100 |
| **PROGRESS** | |
| 4.2% | 89.4 |
| 8.3% | 93.3 |
| 16.7% | 99.0 |
| 32.0% | 100.2 |
| 64.5% | 101.8 |

### B.6 SCALING DATA IMPROVES PROGRESS EVEN MORE THAN VANILLA FULL-FINETUNING

We point to results shown in Table 9 and Fig 7 where scaling data size to higher percentages 32%, 64%, our method outperforms full-finetuning (which uses 100% data) by larger and larger margin (see Rel. Score) as data % is scaled up. **Reason for better scaling performance**- Our method removes redundancy & focuses on most informative samples that the model should learn next - naturally shifts attention toward skills that show strong learning potential, instead of spending excessive effort on skills it already performs well on or are too hard to learn at this instant of time.

### B.7 DETAILS FOR ANALYSIS IN MAIN MANUSCRIPT- HOW DOES THE BENCHMARK DIFFICULTY AND DATA FREQUENCY IMPACT PERFORMANCE?

In this section, we elaborate on details regarding the analysis in Section 4.4, where we analyze the impact of benchmark difficulty and data frequency.

### B.7.1  Details for Benchmark Difficulty Analysis

Here, we provide details regarding the analysis shown in Fig. 8 (a) of main manuscript.

**Quantifying Benchmark Difficulty.**  Prior work has shown that human intuition about task difficulty may not align with a model's difficulty as defined in its feature or hypothesis space (Sachan & Xing, 2016). Therefore, we use the model's own performance as a proxy for determining benchmark difficulty. Specifically, we use the performance of full-dataset fine-tuned LLaVA-v1.5-7B (i.e., Row 0 in Table 1) as reference to determine difficulty of benchmark—benchmarks with higher performance are considered easier. We define **benchmark difficulty** for a given benchmark as $(100 - \text{Performance of full fine-tuned LLaVA-1.5 on Benchmark})/100$. This gives us a difficulty measure for each benchmark normalized between $[0, 1]$ [4].

**Quantifying Performance Improvement.** To isolate the impact of our core contribution—Prioritized Concept Learning (PCL) described in Section 3.2—we measure the performance improvement brought solely by our dynamic sample selection strategy. Specifically, we compute the difference in performance between the full PROGRESS framework (Table 3, Row 7) and the warm-up only model trained prior to applying PCL (Table 3, Row 1). This comparison quantifies the gain attributable to dynamically selecting the most informative samples using our PCL strategy during training.

### B.7.2  Details for Data frequency Analysis

Here, we provide details regarding the analysis shown in Figure 8 (b) of main manuscript.

**Sample Rarity Estimation.**  Our goal is to identify, for each sample in the training dataset, the benchmark it most closely aligns with in terms of skill distribution. Each training sample is assigned to exactly one benchmark—whichever it is closest to—based on similarity in distribution over skills. This allows us to estimate the frequency of training samples aligned with each benchmark, enabling us to quantify how commonly each benchmark's skills are represented in the training data.

**Assignment Procedure.**  To quantify how training samples align with various benchmarks, we use a Gaussian modeling approach. Specifically, we first extract visual and textual features using DINO (for images) and BERT (for questions) and form joint multimodal embeddings as described in Section 3.1—for all samples in training data and each benchmark.

Next, we fit a multivariate Gaussian distribution to each benchmark's embeddings, capturing its mean and covariance to model the underlying skill distribution. Then, for every training sample, we compute the log-likelihood under each benchmark's Gaussian model, reflecting how well the sample fits that benchmark's distribution. Each training sample is then assigned to the benchmark with the highest log-likelihood (refer to Algorithm 1 for full details).

We compute the frequency of training data samples aligned with each benchmark as the proportion of training samples assigned to it:

$$\text{frequency} = \frac{\text{\# matched samples}}{\text{total training samples}}.$$

Finally, we define the rarity as:

$$\text{rarity} = \log(1/\text{frequency}).$$

This formulation enables us to assess how frequently the skills associated with each benchmark appear in the training set (see rarity calculation Algorithim 1).

### B.8  Details for Analysis - What Skills Does the Model Prioritize and When?

In this section, we elaborate on the analysis from Section 4.4 (specifically Fig. 9 in main manuscript), where we investigate which skills the model prioritizes and when during training.

---

[4]For MME, where the full score is not out of 100, we normalize the score by dividing it by the maximum score (2800), the difficulty is computed as $(1 - \text{MME Score}/2800)$

---

**Algorithm 2** Ability Assignment for Clusters

---

**Require:** Training dataset $\mathbb{U} = \{x_1, \ldots, x_N\}$ where $x_i = (I_i, Q_i)$; Clusters $\mathcal{C} = \{C_1, \ldots, C_K\}$; samples from MME benchmark $\mathcal{B} = \{b_1, \ldots, b_M\}$ with ability labels; similarity threshold $\alpha = 0.9$; top-$k$ nearest neighbors ($k = 5$)

1: **Feature Extraction:** Extract DINO features for images and BERT features for questions; concatenate to form joint feature vectors, for all training and benchmark samples. We denote DINO-BERT embedding of $x_i$ as $\phi(x_i)$.

2: **for** each cluster $C_k \in \mathcal{C}$ **do**

3:     **for** each sample $x_i \in C_k$ **do**

4:         $\mathcal{N}_i = \{\text{TopK}(\text{sim}(\phi(x_i), \phi(b_j))_{j=1}^M)\}$ where $\text{sim}(\phi(x_i), \phi(b_j)) = \cos(\phi(x_i), \phi(b_j))$

5:         $\mathcal{N}'_i = \{b_j \in \mathcal{N}_i \mid \text{sim}(\phi(x_i), \phi(b_j)) \geq \alpha \cdot \max_j \text{sim}(\phi(x_i), \phi(b_j))\}$

6:         $\mathcal{A}_i = \{\text{ability}(b_j) \mid b_j \in \mathcal{N}'_i\}$

7:     **end for**

8:     $\mathcal{A}_k = \bigcup_{x_i \in C_k} \mathcal{A}_i$ {Aggregate ability labels from all samples in $C_k$}

9:     $\text{Ability}(C_k) = \text{Mode}(\mathcal{A}_k)$ {Assign the most frequent ability via majority vote}

10: **end for**

11: **return** $\{\text{Ability}(C_1), \ldots, \text{Ability}(C_K)\}$

---

Our goal is to identify the specific ability each skill-cluster—obtained through our concept categorization module described in Section 3.1—represents and track both the number of selected samples and the performance of that skill over time. To do this, we assign each skill cluster in our framework to one of the standardized ability categories defined by the MME benchmark (i.e *count*, *position*, *OCR* etc) which offer interpretable and fine-grained labels covering both perception and cognitive tasks. To determine the dominant ability for each skill-cluster , we use a similarity-based assignment procedure (see Algorithm 2 for details).

We first extract visual and textual features using DINO (for images) and BERT (for questions) and form joint multimodal embeddings as described in Section 3.1—for all samples in training data and MME benchmark dataset.

For each training sample in a given skill-cluster generated by our concept categorization module, we compute its cosine similarity with all samples in the MME benchmark. We identify its top-$K$ nearest neighbors in MME benchmark and retain only those with similarity above a 90% threshold, ensuring that we capture the most aligned MME samples for each training example. MME abilities associated with these filtered neighbors are aggregated for samples in the cluster, and majority voting is applied to assign the most frequent ability to the entire cluster. This process offers a principled way to characterize each skill-cluster's dominant visual-linguistic ability, ensuring robustness through both similarity filtering and voting (refer to Algorithim 2 for more details).

## C ANALYSIS

### C.1 WORD CLOUD VISUALIZATION OF SKILL CLUSTERS

To qualitatively assess the semantic coherence and purity of discovered skill clusters obtained through our concept categorization module (Section 3.1), we generate word clouds by aggregating all questions from all samples assigned to a given cluster. For each cluster, we concatenate all the corresponding questions into a single string and visualize the most frequent words using wordclouds. Note that we remove standard stopwords while plotting the wordclouds.

Figure 12 shows representative word clouds for six clusters. Each cluster exhibits a distinct semantic theme, validating the purity and fine-grained granularity of the automatically discovered clusters and demonstrating the effectiveness of our multimodal concept categorization. For example, cluster (a) object localization and region descriptions, (b) book metadata and genres, (c) pertains to food and nutritional benefits, (d) corresponds to OCR and reference tokens, (e) involves multilingual Japanese text and language prompts, and (f) highlights programming and function-related tasks.

These visualizations demonstrate that our clustering method forms fine-grained, interpretable concept groupings while being fully unsupervised (see Section 3.1)—essential for skill-level tracking and prioritized learning in PROGRESS.

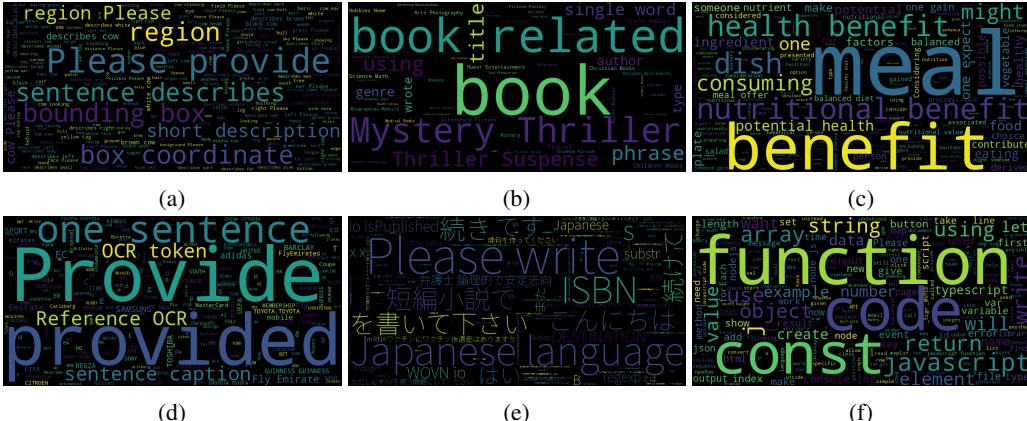

(a)  (b)  (c)

(d)  (e)  (f)

Figure 12: **Word Cloud Visualization of Skill Clusters.** Each subfigure shows the word cloud generated by concatenating all questions within a single skill-cluster discovered by our unsupervised concept categorization module. The clusters exhibit clear semantic themes: (a) object localization and region descriptions, (b) book metadata and genres, (c) food and nutritional benefits, (d) OCR, (e) multilingual Japanese text and language prompts, and (f) programming and function-related instructions. These word clouds highlight the semantic coherence and fine-grained granularity of the automatically discovered clusters, validating their utility for skill-level progress tracking.

## C.2 Skill-level Diversity in Selected Sample Distribution

To better understand the selection behavior across methods, we follow protocol in previous work (Lee et al., 2024) and analyze the number of selected samples from each task in the Vision-Flan-191K dataset. Figure 13 shows the task-wise sample distribution for PROGRESS and several baseline approaches.

We observe that methods relying on single static scoring functions—such as CLIP-Score, EL2N, Perplexity, and Self-Sup—tend to exhibit strong sampling bias, disproportionately selecting from a small subset of tasks while neglecting others. This narrow focus often overlooks important data modes, leading to poor generalization—a limitation also noted in prior work (Lee et al., 2024).

In contrast, PROGRESS maintains a more balanced and diverse sampling profile across tasks, ensuring that a broader range of skills and task types are represented during training. This diversity stems from our skill-driven selection strategy, which tracks learning progress across clusters and samples proportionally using a temperature-controlled distribution.

Overall, by avoiding the pitfalls of static scoring and overfitting to specific high-scoring skills or frequent tasks, our method instead promotes broader and more effective skill acquisition.

## D Limitations

While PROGRESS effectively orders and prioritizes more informative skills, it randomly samples within each selected skill cluster without ranking samples by usefulness. Additionally, the accuracy-based variant incurs extra inference time to compute skill-level progress (see Appendix A), though our loss-based variant avoids this issue. However, overall, PROGRESS outperforms prior approaches while requiring no additional reference VLM and significantly less supervised data.

## E LLM Usage Disclosure

The LLM was primarily used for language refinement rather than content generation—all experimental designs, results, analyses, and scientific contributions are original work by the authors. The LLM assistance was limited to editorial improvements such as fixing grammatical errors, suggesting clearer phrasing for complex technical concepts, and ensuring consistency in terminology throughout the manuscript. No experimental results, mathematical derivations, or scientific claims were generated

by the LLM. All factual statements, citations, and technical content were independently verified by the authors.

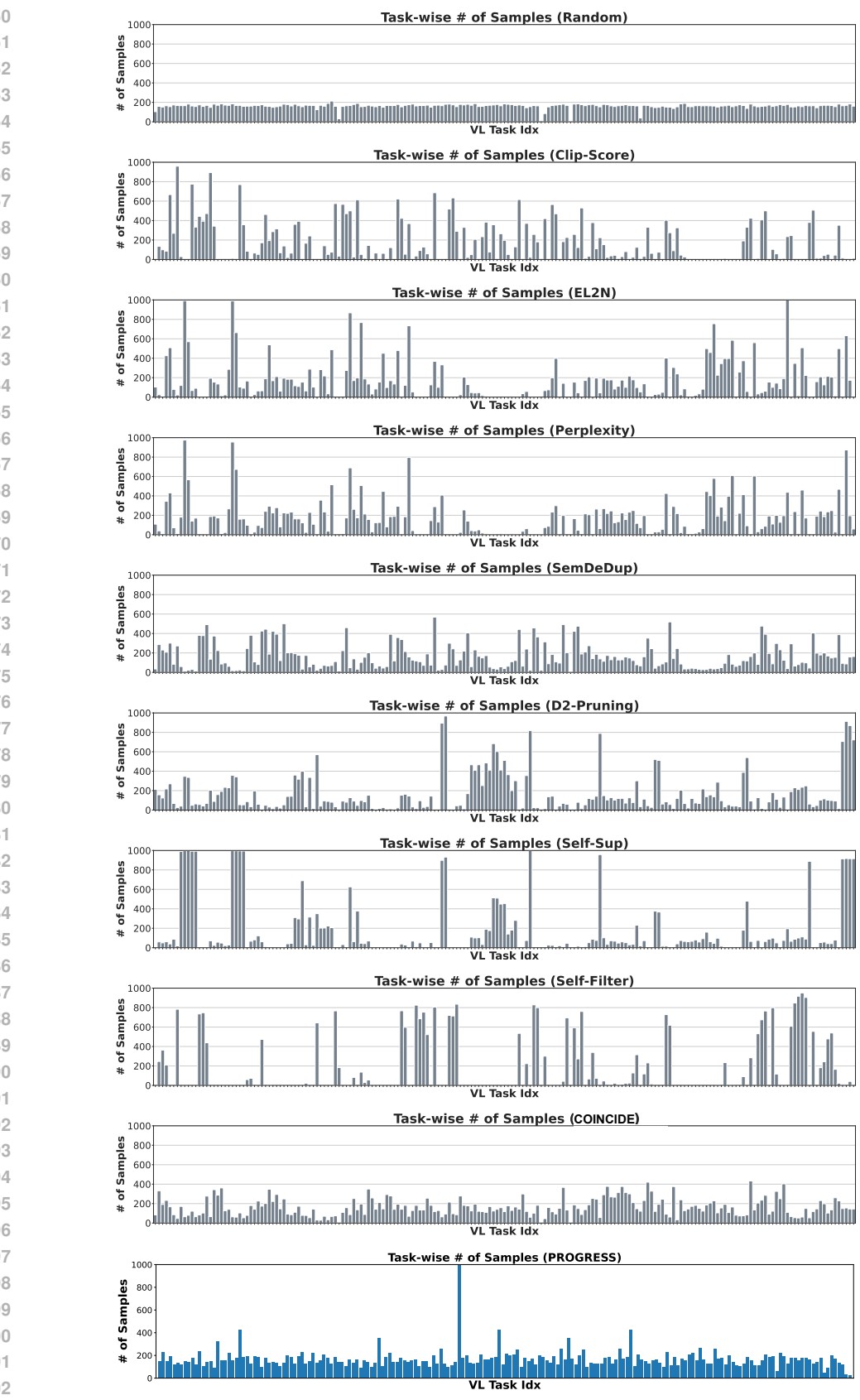

Figure 13: **Task-wise Distribution of Selected Samples.** Number of samples selected (y-axis) from each Vision-Flan-191K task (x-axis) across different methods. While baselines tend to concentrate heavily on a few high-scoring tasks, PROGRESS achieves a more balanced sampling pattern across the task spectrum—highlighting its ability to maintain skill diversity.

