# OpenReview forum: "Learning What Matters: Prioritized Concept Learning via Relative Error-driven Sample Selection"
_ICLR.cc/2026/Conference — ICLR 2026 Conference Withdrawn Submission_

### Official Review · Reviewer_nPzn · 2025-10-28

**Soundness:** 3
**Presentation:** 3
**Contribution:** 2
**Rating:** 4
**Confidence:** 4

**Summary:**

This paper takes the "cluster-then-select" approach to improve data efficiency for fine-tuning MLLMs. The approach is combined with multi-round annotation and curriculum learning to facilitate training. It replaces the tiny reference model used in prior work COINCIDE with specialized visual and textual models for multimodal feature extraction. The method evaluates data using loss/accuracy-based scores and dynamically queries samples in multiple rounds during the fine-tuning process.

**Strengths:**

1. The paper is well-motivated, facilitating MLLM finetuning with multi-round sample selection and curriculum learning.

2. The proposed feature extraction solution, which uses unsupervised specialist models for clustering, presents a compelling and potentially more effective alternative to COINCIDE.

**Weaknesses:**

1. The method's novelty appears incremental as it is heavily built upon the COINCIDE framework. A significant concern is its reliance on an initial 9% data query from the prior method, based on specialist features, for its warm-up phase. This dependency undermines the convincingness of the proposed method's standalone effectiveness, particularly when the main results are reported using only a 20% data subset.

2. The framework has significant limitations for practical application. While multi-round annotation and curriculum learning improve performance, online annotation is often infeasible in modern production pipelines for foundation models. Furthermore, the introduction of a curriculum learning stage, while boosting training performance, can create "gold" data subsets that are less likely to generalize across different model architectures. The added complexity of the framework also introduces more hyperparameters and reduces system robustness, similar to how active learning methods are often criticized for their sensitivity to the data pool, the number of rounds, etc.

3. The use of the term "generalization" is potentially misleading. The experiments effectively apply or validate the proposed method on different architectures and datasets, which is a positive step. However, the paper does not demonstrate that the identified clusters or the selected subsets themselves can be directly generalized to new settings without re-running the entire selection pipeline.

**Questions:**

1. The analysis related to "Skill Prioritization" in Figure 9 is interesting. Is this dynamic a consistent phenomenon observed across other model architectures and datasets, or is it a finding specific to the model and data used in this experiment?

2. In the computational time analysis presented in Table 8, does the reported time include the overhead from the warm-up sample selection stage (the initial 9%)?

3. The number of clusters, K, is set to a very large value in your experiments. While you provide some semantic interpretation for the clusters, were there many clusters resulting from the visual and textual embeddings that were difficult to interpret from a human perception level?

4. How's the performance of full dataset with the proposed curriculum learning scheme?

---

### Official Review · Reviewer_Uu6X · 2025-10-30

**Soundness:** 2
**Presentation:** 2
**Contribution:** 2
**Rating:** 2
**Confidence:** 3

**Summary:**

This paper targets the problem of large-scale datasets and high computational costs in instruction tuning of vision-language models. The authors propose a sample selection method based on the idea of curriculum learning. The hypothesis is that the VLMs can indicate what they can learn from the data.
The proposed method consists of two steps: (1) multimodal concept categorization, using k-means to cluster the vision and text features extracted by DINO and BERT. (2) Prioritized Concept Learning, where the most informative samples are selected based on the improvement in the model's objective.
The experiments show that the proposed method can achieve comparable performance with only up to 20% of the data, thus reducing the computational costs.

**Strengths:**

1. The hypothesis of using the model's own feedback to select informative samples is interesting.
2. The whole idea is quite easy to understand and implement.
3. The experimental results are promising, showing that the method can reduce the amount of data needed.

**Weaknesses:**

1. The motivation is not clear. It is unclear about the skill acquisition and how to prioritize the concept learning. What are the connections between skill and concept? This would be better clarified with concrete examples.
2. This paper seems shallow in VLMs, lacking technical novelty and insights concerning the vision part. The proposed method seems to be a method for all models based on LLMs.
3. The authors claim that tuning the temperature $\tau$ in softmax can balance informativeness and diversity. How do you define diversity here? Informativeness is defined by the improvement of the model's objective, but diversity is not defined clearly.
4. The organization of the paper can be improved, especially the experiment section. Captions of Table 1 can be more concise. And the content is too dense to read.
5. The efficiency analysis is lacking, especially for the sampling time compared to others.

**Questions:**

Please clarify the above weaknesses and answer the following questions:

1. What are the definitions of skill and concept for VLMs in this paper? The number of skills is set to be equal to the number of clusters of clusters. Can you explain this design choice?
2. What is your special design for VLMs here? Can this method be applied to LLMs simply?
3. What is the training efficiency compared to other methods, especially on the sample selection, since you need to use VLMs to evaluate all the samples?
4. Do you try different scoring functions ($\Delta_k$)?
5. What is the definition of accuracy in prioritized concept learning? Since your model does not access the labels, how do you calculate accuracy here?

---

### Official Review · Reviewer_eC7G · 2025-10-30

**Soundness:** 3
**Presentation:** 3
**Contribution:** 2
**Rating:** 4
**Confidence:** 4

**Summary:**

The core contribution of this paper is the proposal of a framework named PROGRESS, which aims to address the issue of massive data and computational resource consumption in the "instruction tuning" process of Vision-Language Models (VLMs).

The core idea of this framework is ingenious: instead of passively receiving all data, it enables the VLM to dynamically and proactively select "what to learn next" during training.

Specifically, it performs the following steps:
- **Track Progress**: At each training stage, the model evaluates its own learning progress (through "self-reflection") across different "skills" or "concepts"—which are automatically identified via unsupervised clustering.
- **Prioritize Selection**: The model prioritizes data samples corresponding to skills where its learning progress is the fastest (i.e., the most significant decline in "relative error").
- **Intelligent Filtering**: This strategy ensures the model focuses on content in its "Zone of Proximal Development" (ZPD)—content that (a) it has not yet fully mastered, but (b) is not "too difficult to learn."


Compared with traditional methods, the main advantages of PROGRESS are:
- **Data Efficiency**: It achieves 99-100% of the performance of training on the full dataset while using only 16-20% of the annotated data.
- **Annotation Efficiency**: It does not require expensive pre-annotation of answers for the entire dataset; instead, it queries (or generates) answers for a sample "on demand" only when it decides to learn that sample.
- **Computational Efficiency**: It does not require an additional, pre-trained auxiliary VLM to "guide" learning, nor does it rely on computationally intensive gradients for sample selection.
- **Strong Generalizability**: Experiments show that this method not only delivers good performance but also generalizes well to VLMs with different architectures and larger scales.

**Strengths:**

This paper proposes PROGRESS, whose core advantage lies in its extremely high data, annotation, and computational efficiency. It adopts a dynamic strategy inspired by curriculum learning, enabling the model to proactively select samples during training based on its own evolving needs. The method tracks the learning progress of different skills and prioritizes samples where the model achieves the "fastest progress," thereby effectively controlling the order in which skills are acquired.

This strategy is not only practical but also cost-effective: it eliminates the need for expensive pre-existing global annotations and only queries answers "on demand." Meanwhile, it avoids reliance on additional auxiliary VLMs (Vision-Language Models) and the massive computational load of gradients. Ultimately, PROGRESS achieves performance close to full-data training using only 16-20% of supervised data, outperforms existing SOTA (State-of-the-Art) methods on multiple benchmarks, and demonstrates strong cross-architecturegeneralization and scalability to larger models.

**Weaknesses:**

1. This method requires a "warmup" phase (using 9% of the data in the paper), which is claimed to enable the model to obtain a "reliable" initial performance evaluation. The problem is that it can be seen from Appendix Figure 10(a) that the warmup ratio is extremely critical. A 9% ratio yields the best results, while ratios of 3% or 12% both lead to a significant decline in performance. This means the "initial state" of the model greatly affects the subsequent calculation of "relative progress". This is yet another hyperparameter that requires fine-tuning, making the method less robust.

2. Potential neglect of "fundamental but slow-to-learn" skills
The core of PROGRESS is to prioritize learning skills with the "fastest progress" (highest $\Delta_k$). The question is: is this strategy always optimal?
Fundamental skill trap: Imagine a "fundamental skill" (such as understanding spatial orientation). It may be difficult to learn and progress slowly (thus having a low $\Delta_k$), but it is a key prerequisite for subsequent learning of advanced skills like "localization" and "counting". The PROGRESS framework may keep delaying the learning of this fundamental skill due to its low $\Delta_k$, and instead turn to learning those "fast-progressing but less important" skills. This causes the model to "hit a wall" in the later stages and fail to master more complex capabilities.

3. Lack of explicit monitoring for "skill forgetting"
This framework only focuses on "relative progress", i.e., the situation where $\Delta_k$ is positive. The problem is that it lacks a clear mechanism to address performance degradation (Catastrophic Forgetting). When the model is striving to learn Skill B (with fast progress), it is entirely possible that this leads to a decline in its performance on Skill A, which it has already mastered before. Since $\Delta_k$ only looks at "progress", the framework may not immediately notice that Skill A is being forgotten. It may only be re-identified as a "fast-progressing" skill at some stage in the future, but this is highly inefficient.

**Questions:**

I understand that the first $\Delta_k$  is derived from the gap between the result after warm-up and the result after retraining on one batch of data. But is the selection of this first batch of data random?

---

### Official Review · Reviewer_9y1S · 2025-10-31

**Soundness:** 3
**Presentation:** 3
**Contribution:** 3
**Rating:** 6
**Confidence:** 3

**Summary:**

The paper presents PROGRESS, a data and compute-efficient framework for visual instruction tuning that learns from an initially unlabeled pool. It first clusters image–question pairs into semantically coherent “skills” using self-supervised image and text features. Training then prioritizes the clusters that exhibit the most recent improvement, sampling a small number of examples from them for on-demand annotation, which yields an adaptive curriculum. This removes reliance on auxiliary VLMs for selection and focuses effort where the model is currently learning fastest. Across several backbones and benchmarks, the approach reaches near full-data performance with a fraction of labeled data, supported by ablations on sampling temperature, warm-up size, and selection gaps. The paper also notes practical trade-offs and limitations, such as the use of a text-only judge to estimate progress and sensitivity to the warm-up phase.

**Strengths:**

- Achieves near–full-data performance with a small labeled subset; strong results hold across multiple model sizes and benchmarks.

- Shows that progress-driven sampling at a cluster/“skill” level with a simple temperature-controlled softmax yields a stable, effective curriculum.

- Extensive Evaluation across diverse benchmarks, model families, budget scales, and wall-clock comparisons.

- Thorough pipeline, tables, and ablation studies; appendices provide  hyper-parameters and implementation details that aid reproducibility.

**Weaknesses:**

**Characterization of skills**: The approach implicitly treats DINO+BERT clusters as skills, but the manuscript provides limited evidence that these clusters correspond to meaningful or stable competencies. A more concrete analysis such as semantic labeling of clusters, stability across feature backbones and seeds, and sensitivity to the number of clusters, would help readers understand what is being learned and whether the method targets distinct capabilities rather than surface correlations. (+ What skills are prioritized during training? in Section 4.4 are hard to interpret and does not show a clear trend as the authors argue.)

**Mechanistic explanation of gains**: Several results show subsets matching or exceeding full-data fine-tuning, but the paper does not explain why this happens. (It adds empirical analysis yet does not directly explain why it often surpasses full data fine-tuning which should be more advantageous in general.) Possible factors include redundancy and noise in the full dataset, conflicting instructions across sources, or interference that curriculum mitigates. Without diagnostics, it is unclear whether the method works because it identifies the right “skills,” because it filters low-quality/duplicated data, or simply because it schedules training more gently. Moreover, the sampling rule (combining loss or accuracy improvement with a softmax diversity heuristic) remains somewaht heuristic and lacks theoretical grounding or ablation against alternative criteria (e.g., gradient-based, uncertainty-based, or generalization-driven sampling). This raises the possibility that the method’s success could be suboptimal and that other heuristics could yield similar or even stronger effects under the same framework.

**Limited Novelty**: The method reads as a straightforward recombination of prior ideas, differing only modestly from COINCIDE and standard active-learning strategies. To strengthen the academic contribution, the paper should explain why the approach works, clarifying the mechanism by which cluster-level progress helps and delineate when and how active-learning principles transfer to vision–language instruction tuning/agents.

**Questions:**

- Could you clearly articulate the paper’s unique contribution relative to COINCIDE, curriculum/self-paced learning, and active learning? Specifically, what does cluster-level progress add beyond uncertainty/loss-based sampling or curriculum without clustering, and under what conditions is it necessary?

- Please provide evidence that clusters correspond to coherent capabilities rather than topical groupings. For example: a small human audit that tags 100–200 clusters with interpretable labels; stability metrics across feature backbones (e.g., DINO vs. CLIP), and K-values, etc. and a mapping from clusters to ability categories showing that frequently selected clusters drive targeted gains.

- Do you have diagnostics or structured reason that well-explains why subsets can equal or **surpass** full-data training?

---

### Official Review · Reviewer_Dht8 · 2025-11-10

**Soundness:** 3
**Presentation:** 3
**Contribution:** 2
**Rating:** 4
**Confidence:** 4

**Summary:**

This paper introduces PROGRESS, a data-efficient framework designed to address the high costs associated with the instruction-tuning of Vision-Language Models (VLMs). The core mechanism involves partitioning samples into "skill" clusters via unsupervised learning and dynamically tracking the model's learning progress on these clusters during training. The framework then prioritizes and annotates samples that yield the highest "relative performance improvement," thereby guiding the model toward an efficient learning trajectory. Experimental results demonstrate that with only 16-20% of the labeled data, this method can achieve performance comparable to, and in some cases surpassing, full-finetuning on 100% of the data.

**Strengths:**

### Significance and Practicality
The paper addresses the critical bottleneck of high computational and annotation costs in VLM training. This is a highly relevant and valuable research direction that is crucial for advancing research and accessibility within the community.

### Pragmatic Framework Design
The method avoids reliance on additional large-scale auxiliary models or a fully pre-annotated dataset. Its "on-demand" annotation strategy makes it more feasible and scalable for real-world deployment.

### Solid Empirical Validation
The authors have conducted extensive experiments across multiple model architectures, datasets, and a diverse suite of up to 14 downstream benchmarks, providing strong empirical evidence for the method's effectiveness and generalization capabilities.

**Weaknesses:**

### The Validity of the "Skill" Definition Requires Further Substantiation
The framework's entire premise rests on the assumption that unsupervised clustering effectively partitions data into meaningful, distinct "skills." However, the case studies in Figure 3 (e.g., the "Grounding" skill) show that samples within a cluster share nearly identical question phrasing. This raises a significant concern that the clustering may be predominantly driven by textual patterns from BERT rather than capturing a more abstract, multimodal cognitive skill. Consequently, it is unclear whether the model is learning a generalizable "skill" or simply becoming more efficient at responding to specific "textual templates."

### Potential Computational Bottleneck in the Self-Evaluation Phase
The paper highlights the framework's efficiency in terms of total training and annotation time. However, PROGRESS introduces a "stop-and-go" workflow that requires periodically pausing training to perform large-scale inference on sample pools from multiple skill clusters for evaluation. This periodic inference overhead could itself become a significant efficiency bottleneck, especially as model and dataset sizes scale, potentially negating the training time saved. The paper lacks a thorough discussion of this potential scalability issue.

### Limited Novelty of the Core Selection Mechanism
The core idea of selecting samples based on "relative improvement" draws heavily from established principles in Curriculum Learning and Self-Paced Learning. As such, the work reads more like a successful application and engineering of these ideas in the VLM domain rather than the proposal of a fundamentally new learning theory.

### Unclear Dependency on the Warm-up Strategy
The framework's successful initialization relies on a sophisticated warm-up stage that uses nearly half of the data budget (9% out of 20%) and adopts a complex selection strategy from prior work. While the paper's ablation study demonstrates that the subsequent dynamic selection stage is essential, it does not fully clarify the framework's dependency on this specific warm-up strategy. It remains uncertain whether the framework would achieve similarly impressive results if initialized with a simpler warm-up method (e.g., random sampling).

**Questions:**

1. Regarding the Generalization of "Skill" Clusters: Figure 3 suggests that clusters may be strongly influenced by textual patterns, as samples share similar phrasings. To validate that these clusters represent abstract skills, could you provide examples where samples targeting the same underlying task (e.g., object localization) but using diverse linguistic prompts are correctly grouped into a single cluster?

2. Regarding the Practical Overhead of Self-Evaluation: Could you provide a more detailed time/compute cost analysis that specifically isolates the periodic "pause-and-evaluate" phase? How do you see this overhead scaling with much larger models (e.g., 100B+ parameters) and datasets (e.g., tens of millions of samples)? Is there a tipping point where this evaluation overhead becomes impractical?

3. Regarding the Explanation for Surpassing Full-Finetuning: For the impressive results on tasks like ChartQA, where using only 20% of the data surpassed the 100% data baseline, could you offer a deeper analysis? What are the characteristics of the 80% of samples that PROGRESS discarded? Is there evidence to suggest they are low-quality, noisy, redundant, or even "poisonous" for learning these specific advanced skills?

4. Regarding the Robustness of the Warm-up Strategy: To better understand the robustness of your framework, have you conducted a comparative experiment where the model is warmed up using the same amount of data (9% of the total) but selected via simple random sampling, before applying the PROGRESS dynamic selection? This experiment would more clearly reveal the extent to which the framework's success depends on a sophisticated "head start" versus the merits of the subsequent dynamic selection process alone.

---

### Note · Authors · 2025-11-14

I have read and agree with the venue's withdrawal policy on behalf of myself and my co-authors.